# Pretectal neurons control hunting behaviour

**Paride Antinucci[1], Mónica Folgueira[2,3], Isaac H Bianco[1]\***

[1]Department of Neuroscience, Physiology & Pharmacology, UCL, London, United Kingdom; [2]Department of Biology, Faculty of Sciences, University of A Coruña, A Coruña, Spain; [3]Centro de Investigaciones Científicas Avanzadas (CICA), University of A Coruña, A Coruña, Spain

**Abstract** For many species, hunting is an innate behaviour that is crucial for survival, yet the circuits that control predatory action sequences are poorly understood. We used larval zebrafish to identify a population of pretectal neurons that control hunting. By combining calcium imaging with a virtual hunting assay, we identified a discrete pretectal region that is selectively active when animals initiate hunting. Targeted genetic labelling allowed us to examine the function and morphology of individual cells and identify two classes of pretectal neuron that project to ipsilateral optic tectum or the contralateral tegmentum. Optogenetic stimulation of single neurons of either class was able to induce sustained hunting sequences, in the absence of prey. Furthermore, laser ablation of these neurons impaired prey-catching and prevented induction of hunting by optogenetic stimulation of the anterior-ventral tectum. We propose that this specific population of pretectal neurons functions as a command system to induce predatory behaviour.

DOI: https://doi.org/10.7554/eLife.48114.001

**\*For correspondence:**
i.bianco@ucl.ac.uk

**Competing interests:** The authors declare that no competing interests exist.

## Introduction

In response to sensory information and internal states, animals select specific actions from a repertoire of options and produce adaptive behavioural programmes. Neuroethological studies in a variety of species have pinpointed brain regions, and identified neurons, that specifically promote particular behaviours ranging from the courtship songs of fruit flies (*von Philipsborn et al., 2011*; *Inagaki et al., 2014*) to parental behaviours of mice (*Kohl et al., 2018*). Identifying the neural circuits that control specific behaviours, as well as modulatory systems that influence if and how behaviours are performed, will shed light on the neural mechanisms of decision making, action selection and motor sequence generation.

Prey catching is an innate, complex behaviour that is crucial for survival (*Sillar et al., 2016*). In various species, hunting responses can be evoked by prey-like stimuli, defined by specific conjunctions of sensory features (*Ewert, 1997*; *Anjum et al., 2006*; *Bianco and Engert, 2015*), and predatory behaviour is modulated by internal state variables including associative learning and feeding drive (*Ewert et al., 2001*; *Jordi et al., 2015*). Several brain regions show activity during hunting and are expected to subserve neural functions including prey detection and localisation, control of pursuit, capture and consummatory actions, and motivation (e.g. *Comoli et al., 2005*). Although electrical stimulation of brain regions, including the optic tectum, can evoke hunting actions (*Ewert, 1970*; *Bels et al., 2012*) and recent studies in rodents have identified circuits that motivate predatory behaviour (*Han et al., 2017*; *Li et al., 2018*; *Park et al., 2018*), premotor circuits that directly control vertebrate hunting have yet to be identified.

In this study, we sought to identify neurons that control predatory behaviour, using larval zebrafish as a vertebrate model system. In larval zebrafish, hunting is an innate, visually guided behaviour, which involves a sequence of specialised oculomotor and locomotor actions. A defining

**eLife digest** Hunting is an innate behaviour that relies on predators executing a precise set of actions to identify, approach, target, and strike prey. In vertebrates (animals with a backbone), the identity of the brain circuits that trigger hunting is still unclear. These networks are thought to link sensory perception (seeing prey) with a specialised action (starting an attack).

Larval zebrafish are a good model in which to study these circuits, because they have a tiny, transparent brain where neurons can be observed in real time. In addition, it is expected that the networks that control hunting in this species will be preserved across other vertebrates.

To discover these networks, Antinucci et al. genetically engineered zebrafish larvae so that their brain cells would 'glow' when they became active. This revealed which individual brain cells would turn on when zebrafish started to hunt. These neurons were in a part of the brain called the pretectum, which receives visual information about prey from the eye.

Next, Antinucci et al. harnessed a technique called optogenetics to artificially turn on these brain cells in the pretectum, which caused the fish to start hunting even when prey was absent. In fact, stimulating just one pretectal cell was enough to trigger the behaviour. Conversely, killing pretectal brain cells using precise laser surgery hindered hunting in zebrafish exposed to prey.

These experiments suggest that pretectal brain cells act like a command centre that controls hunting. They likely play a decision-making role, determining when animals do and do not respond to events in their surroundings. Similar neurons likely control other types of behaviour. Understanding how these circuits work at the cellular level in zebrafish may help scientist study them in other organisms, such as humans.

DOI: https://doi.org/10.7554/eLife.48114.002

characteristic is that larvae initiate hunting by rapidly converging their eyes, which substantially increases their binocular visual field (*Bianco et al., 2011*). A high vergence angle is maintained during prey pursuit, which entails a sequence of discrete orienting turns and approach swims, which culminate in binocular fixation of prey followed by a kinematically distinct capture swim (*Borla et al., 2002*; *McElligott and O'malley, 2005*; *Bianco et al., 2011*; *Patterson et al., 2013*; *Trivedi and Bollmann, 2013*; *Marques et al., 2018*). Neural activity associated with prey-like visual cues has been detected in the axon terminals of a specific class of retinal ganglion cell (RGC), which terminate in retinal arborisation field 7 (AF7) in the pretectum (*Semmelhack et al., 2014*). Prey-responsive pretectal cells have also been described (*Muto et al., 2017*) as well as highly prey-selective feature-analysing neurons in the optic tectum (OT) that display non-linear mixed selectivity for conjunctions of visual features (*Bianco and Engert, 2015*). Premotor activity in localised tectal assemblies immediately precedes execution of hunting responses (*Bianco and Engert, 2015*) and optogenetic stimulation of the anterior-ventral tectal region can induce hunting-like behaviour (*Fajardo et al., 2013*). Finally, ablation of RGC input to either AF7 or OT substantially impairs hunting (*Gahtan et al., 2005*; *Semmelhack et al., 2014*).

Based on this evidence, we hypothesised that neural circuits controlling the induction of hunting might be located in the vicinity of AF7 or OT. Our experimental requirements for identifying such neurons were (1) that they display neural activity specifically related to execution of hunting behaviour, rather than visual detection of prey, and (2) that direct stimulation of such neurons would induce naturalistic predatory behaviour in the absence of prey. First, we used 2-photon calcium imaging paired with a virtual hunting assay and identified a high density of neurons in the AF7-pretectal region that were specifically recruited when larvae initiated hunting behaviour. We identified a transgenic line that labelled these neurons and found two morphological classes: One projects ipsilaterally to the optic tectum and the second extends long-range projections to midbrain oculomotor nuclei, the nucleus of the medial longitudinal fasciculus and the contralateral hindbrain tegmentum. Remarkably, optogenetic stimulation of single pretectal neurons evoked hunting-like behaviour in the absence of prey. Pretectal projection neurons of either class could evoke hunting routines with naturalistic oculomotor and locomotor kinematics but opposite directional biases. Finally, laser-ablation of the pretectal population impaired hunting of live prey and prevented induction of hunting by optogenetic stimulation of the rostral tectum. In sum, we propose that this specific population of

pretectal neurons functions downstream of prey perception to directly control execution of predatory behaviour.

## Results

### Neural activity associated with hunting initiation

To identify and distinguish neurons with activity related to prey perception and initiation of hunting behaviour, we performed 2-photon calcium imaging while larval zebrafish engaged in a virtual hunting assay (*Figure 1A*) (*Bianco and Engert, 2015*). Transgenic *elavl3:H2B-GCaMP6s;atoh7:gapRFP* larvae (6–7 dpf, N = 8) were partially restrained in agarose gel, but with their eyes and tail free to move, and were presented with a range of visual cues including small moving prey-like spots, which evoke naturalistic hunting responses (*Figure 1B*) (*Bianco et al., 2011*; *Bianco and Engert, 2015*). We imaged a volume that encompassed the majority of the primary retinorecipient sites [arborisation fields (AFs) 2–10] as well as surrounding brain regions including pretectum and OT (310 × 310 × 100 μm volume; *Figure 1C* and *Video 1*). Eye and tail kinematics were tracked online, allowing automated detection of hunting responses. These are defined by saccadic convergence of the eyes – an oculomotor behaviour specific to hunting initiation and frequently coincident with lateralised tail movements (91.6 ± 4.9% of convergent saccades paired with tail movement; eye-tail latency 11.6 ± 5.2 ms, mean ± SD; *Figure 1D* and *Figure 1—figure supplement 1A,B*) (*Bianco et al., 2011*; *Patterson et al., 2013*; *Trivedi and Bollmann, 2013*; *Bianco and Engert, 2015*). Larvae preferentially responded to small, dark, moving spots and hunting was initiated most frequently once the stimulus had crossed the midline axis and was moving in a nose-tail direction (*Figure 1E,F* and *Figure 1—figure supplement 1C*) (*Bianco and Engert, 2015*).

To define groups of neurons with consistent functional properties related to the first stages of hunting behaviour, we first computed, for every cell, a visuomotor vector (VMV) that quantified its sensory and motor-related activity (*Figure 1G* and Materials and methods). Each VMV described (a) mean GCaMP fluorescence responses to each of the ten visual cues during non-response trials, in which larvae did not release hunting behaviour, and (b) coefficients from multilinear regression of fluorescent timeseries data on a set of motor predictors (derived from eye and tail kinematics; *Figure 1—figure supplement 1E* and *Supplementary file 1*). Three motor predictors indicated convergent saccades associated with leftwards, rightwards or symmetrical/no tail movements (labelled 'Conv tail L', 'Conv tail R' and 'Conv tail sym', respectively). Next, we used an unsupervised clustering procedure to identify consistent sensorimotor tuning profiles. A correlation-based agglomerative hierarchical clustering algorithm performed initial clustering of VMVs from cells with either high visually evoked activity or that were well modelled in terms of motor variables (comprising ~10% of neurons; *Figure 1—figure supplement 1D* and Materials and methods). The centroids of the resultant cluster 'seeds' defined a set of functional archetypes (*Figure 1—figure supplement 1F*) and subsequently, all remaining neurons were assigned to the cluster with the closest centroid within a threshold distance limit (Pearson's $r \geq 0.7$, VMVs below threshold remained unassigned). This approach allowed us to classify more than 50% of imaged neurons (93,054 out of 181,123 cells) into 36 clusters with homogenous functional properties (*Figure 1H* and *Figure 1—figure supplement 1G–I*).

This analysis identified neurons that were recruited during hunting initiation. Specifically, four clusters showed activity highly correlated with eye convergence events (clusters 25–28; *Figure 1I*) yet exhibited little activity in response to visual cues (including prey-like moving spots; *Figure 1J* and *Figure 1—figure supplement 2A,B*). In two of these clusters, activity was selective for the direction of motor responses. Thus, cluster 26 was selective for convergent saccades associated with leftwards turns and cluster 28 was tuned to rightwards hunting responses. By contrast, the other two clusters did not show selectivity for the direction of tail movements during hunting initiation (cluster 25, associated with symmetrical/no tail movement and 27, responsive to motion in either direction). Other functional clusters comprised visually responsive neurons that were selectively activated by small prey-like moving spots ('prey-responsive' clusters 1–6; *Figure 1H–J* and *Figure 1—figure supplement 2A*), but displayed little motor-related activity.

We computed a 'hunting index' (HIx) for individual neurons as a direct means to distinguish neural activity associated with hunting initiation from 'sensory' activity evoked by prey-like visual stimuli. Briefly, for each hunting response, GCaMP fluorescence in a time window (±1 s) surrounding the

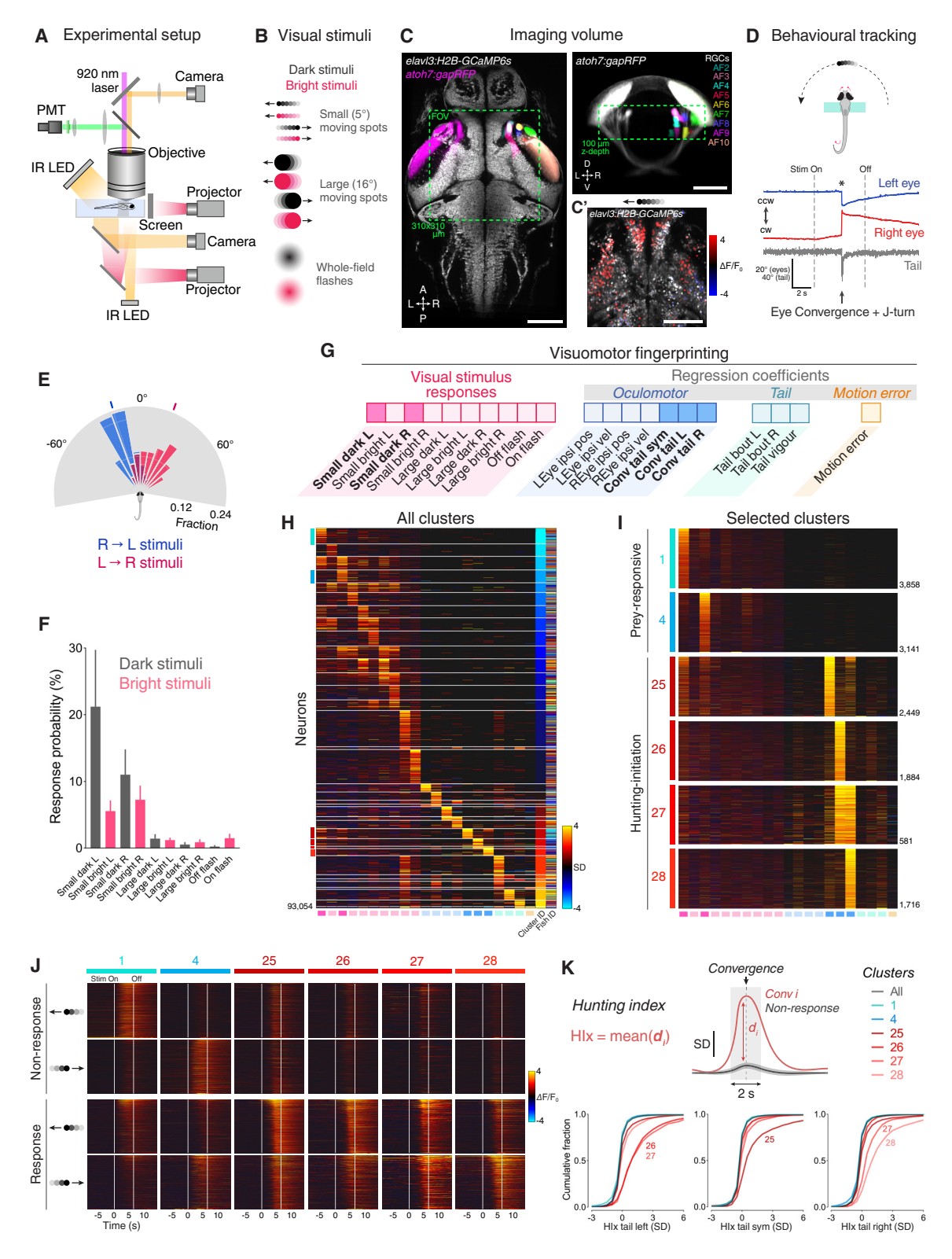

**Figure 1.** Neural activity associated with hunting. (**A**) 2-photon GCaMP imaging combined with behavioural tracking during virtual hunting behaviour (see Materials and methods). (**B**) Schematic of visual stimuli. (**C**) *elavl3:H2B-GCaMP6s;atoh7:gapRFP* reference brain showing imaging volume (green box), which encompassed most retinal arborisation fields (AF2–10). In the right hemisphere, RFP has been pseudo-coloured to demarcate specific AFs. (**C'**) Example of neuronal activity (ΔF/F₀) within one focal plane in response to a dark, leftwards moving prey-like spot (mean activity over eight

*Figure 1 continued on next page*

*Figure 1 continued*

presentations) overlaid onto anatomical image (grey). (D) Example of behavioural tracking data indicating hunting initiation (eye convergence and leftwards J-turn) in response to a dark, leftwards moving prey-like spot. Asterisk indicates time of convergent saccade. cw, clockwise; ccw, counter-clockwise. (E) Distribution of spot locations at time of convergent saccade. Ticks indicates median location for leftwards (blue, –18.13°, N = 162 events in eight fish) and rightwards (red, 22.10°, N = 122 events) moving spots. (F) Hunting response probability (mean + SEM, N = 8 fish) across visual stimuli. (G) Schematic of the visuomotor vector (VMV) generated for each neuron. (H) VMVs of all clustered neurons (N = 93,054 neurons from eight fish). Within each cluster, neurons are ordered according to decreasing correlation with the cluster seed centroid (mean VMV). Coloured lines on the left highlight hunting-related clusters (prey-responsive clusters in blue, hunting-initiation clusters in red). (I) Enlargement showing VMVs of selected hunting-related clusters (1, 4, 25–28). Number of cells in each cluster is shown on right. (J) Stimulus-aligned activity during non-response (top) and response (bottom) trials for neurons in selected clusters (indicated top). (K) Hunting Index (HIx). Top schematic indicates how HIx is computed from calcium signals and bottom shows distribution of HIx scores for selected clusters. Scale bars, 100 µm. A, anterior; D, dorsal; L, leftwards; P, posterior; R, rightwards; V, ventral; Sym, symmetric. See also *Figure 1—figure supplements 1* and *2* and *Video 1*.

DOI: https://doi.org/10.7554/eLife.48114.003

The following source data and figure supplements are available for figure 1:

**Source data 1.** Source data for *Figure 1*.

DOI: https://doi.org/10.7554/eLife.48114.006

**Figure supplement 1.** Behavioural and clustering analyses.

DOI: https://doi.org/10.7554/eLife.48114.004

**Figure supplement 2.** Stimulus and motor-triggered calcium responses.

DOI: https://doi.org/10.7554/eLife.48114.005

convergent saccade was compared to activity at the same time in non-response trials during which the same visual stimulus was presented (*Figure 1K*, top). The mean of these difference measures across all response events represents the HIx score for the cell and quantifies neural activity attributable to hunting initiation while accounting for any visually evoked response. To account for directional tuning, we separately computed HIx for hunting responses paired with leftwards, rightwards or symmetrical/no tail movements. This analysis revealed that neurons in clusters 25–28 showed considerably higher HIx scores than other cells, including those in prey-responsive clusters (1 and 4, *Figure 1K*). Tail directional preferences were consistent with those determined by regression modelling.

Overall, our functional analyses identified four clusters of neurons with activity specifically associated with the specialised motor outputs that characterise initiation of hunting behaviour and showed little activity in response to prey-like visual cues. Thus, we will refer to these as 'hunting-initiation' clusters.

## Pretectal neurons are recruited during hunting initiation

A high density of neurons in hunting-initiation clusters were located in the pretectum. We showed this by registering volumetric imaging data to a reference brain atlas ('ZBB' and a high-resolution *elavl3:H2B-GCaMP6s* volume, see Materials and methods and *Video 1*) (*Marquart et al., 2015*; *Marquart et al., 2017*). Neurons within functionally defined clusters showed distinct anatomical distributions (*Figure 2—figure supplement 1*). Notably, a high density of neurons belonging to hunting-initiation clusters were found in pretectal regions in the immediate vicinity of AF7 (AF7-pretectum), just anterior to the rostral pole of the optic tectum (*Figure 2A,B* and *Figure 2—figure supplement 2*). Neurons selective for the direction of hunting-related tail motion (clusters 26 and 28) showed lateralised, mirror-symmetric distributions, with a larger fraction of cells located on the side of the brain contralateral to the direction of preferred tail movement (*Figure 2A*, right panel). Specifically, cluster 26, which is tuned to eye convergence associated with leftwards tail movement, had a higher

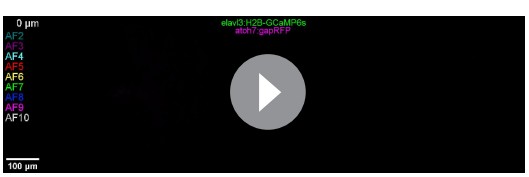

**Video 1.** Z-stack of transgenic line used for calcium imaging with annotated RGC arborisation fields. Imaging volume (z-stack) of 6 dpf *elavl3:H2B-GCaMP6s; atoh7:gapRFP* brain (mean of N = 3 fish) with labelled RGC arborisation fields (AFs). The green channel shows the *elavl3:H2B-GCaMP6s* reference brain used for anatomical registration. Related to *Figure 1*.

DOI: https://doi.org/10.7554/eLife.48114.007

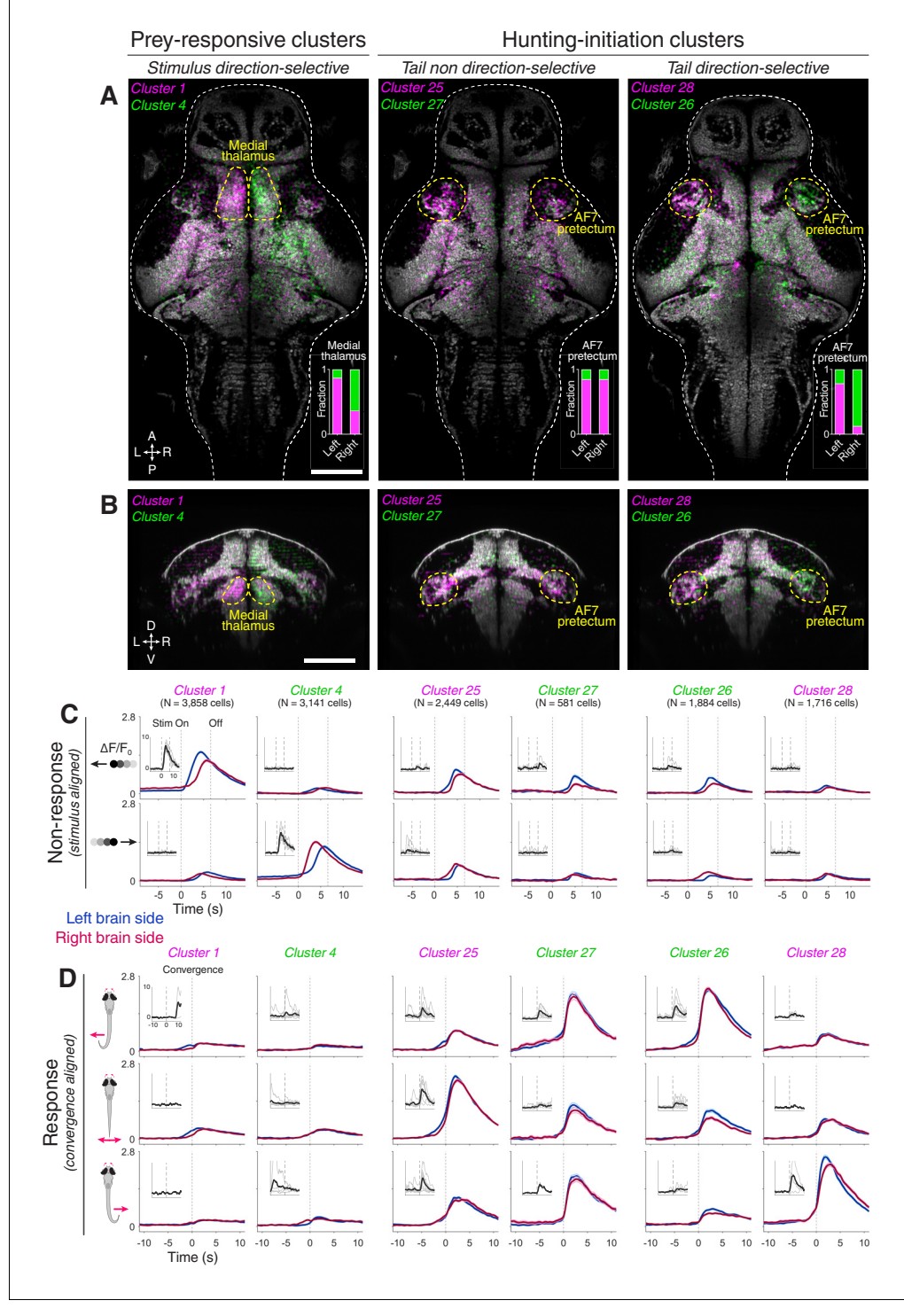

**Figure 2.** AF7-pretectum contains a high density of hunting initiation neurons. (**A**) Anatomical maps of prey-responsive clusters (left) and hunting-initiation clusters (middle and right). Images show dorsal views of intensity sum projections of all neuronal masks in each cluster after registration to the *elavl3:H2B-GCaMP6s* reference brain (grey). Insets show fraction of neurons in left and right AF7-pretectum or medial thalamus belonging to specified clusters. (**B**) Ventro-dorsal cross-section views of anatomical maps. (**C**) Visual stimulus-aligned activity during non-response trials for prey-responsive clusters (left) and hunting-initiation clusters (middle and right; mean ± SEM). Activity is displayed separately for left (blue) and right (red) hemisphere neurons in each cluster. Insets show single-trial responses for a single example cell from each cluster (mean as thick line). (**D**) Activity aligned to convergent saccades associated with leftwards (top), rightwards (bottom), or symmetrical/no tail movements

*Figure 2 continued on next page*

*Figure 2 continued*

(middle). Activity during both spontaneous and visually evoked events was used. Scale bars, 100 μm. A, anterior; D, dorsal; L, left; P, posterior; R, right; V, ventral; Stim, stimulus.

DOI: https://doi.org/10.7554/eLife.48114.008

The following figure supplements are available for figure 2:

**Figure supplement 1.** Anatomical maps of clusters.

DOI: https://doi.org/10.7554/eLife.48114.009

**Figure supplement 2.** Anatomical locations of clustered neurons.

DOI: https://doi.org/10.7554/eLife.48114.010

---

density of cells in the right AF7-pretectum, and vice versa for cluster 28. Hunting-initiation clusters that were agnostic to tail direction (clusters 25 and 27) showed largely symmetric anatomical distributions (*Figure 2A*, middle panel). Neurons belonging to prey-responsive clusters were also found in AF7-pretectum as well as in the medial thalamus, where direction-selective neurons showed highly lateralised distributions (*Figure 2A,B*, left panel).

To further examine visuomotor tuning, we computed visual stimulus-aligned and convergence-aligned activity profiles separately for left and right hemisphere neurons. This confirmed that prey-responsive neurons in clusters 1 and 4 showed direction-selective activity in response to small dark moving spots, but minimal activity associated with convergent saccades (*Figure 2C,D*, left columns). In contrast, hunting-initiation neurons (clusters 25–28) showed weak visual responses – as shown by moving spot-triggered activity during non-response trials – but substantial activity triggered on hunting initiation. For clusters 26 and 28, neurons showed stronger activation when convergent saccades were paired with contraversive left and right-sided turns, respectively (*Figure 2C,D*, right columns).

In summary, we identified populations of neurons in AF7-pretectum that are active in association with two distinct components of hunting – visual responses to prey and initiation of predatory behaviour. We subsequently examined whether pretectal cells with hunting-initiation activity are directly involved in inducing hunting behaviour.

## AF7-pretectal neurons are labelled by the *KalTA4u508* transgene

To characterise the connectivity and function of AF7-pretectal neurons with hunting-initiation activity, we inspected the expression patterns of a range of transgenic driver lines and identified a transgene, *KalTA4u508*, which labels neurons in the AF7-pretectal region (*Figure 3A–C*). Additionally, *KalTA4u508* labels populations of cells within the midbrain, hindbrain, sensory ganglia and spinal cord as well as non-neural tissues (*Figure 3—figure supplement 1A*). However, anatomical registration of *KalTA4u508;UAS:mCherry* volumetric data to the reference atlas revealed a high density of labelled somata in AF7-pretectum, overlapping with the locations of hunting-initiation clusters (*Figure 3A,B,I*). We generated *KalTA4u508;UAS:RFP;atoh7:GFP* larvae to visualise GFP-labelled RGC axon terminals in AF7, and observed that a subset of *KalTA4u508*-expressing neurons extend dendritic arbours that directly juxtapose RGC terminals (N = 4 fish; *Figure 3C,D*). In brain sections from adult (3 month old) *KalTA4u508;UAS:GCaMP6f;atoh7:gapRFP* fish, labelled neurons in the pretectum were very sparse and could be identified only in the accessory pretectal nucleus (APN; N = 8 somata from four fish; *Figure 3E*). This suggests that at least a subset of *KalTA4u508*-expressing neurons reside in a region of the larval AF7-pretectum corresponding to the adult APN.

Next, we sought to examine the morphology of pretectal *KalTA4u508* neurons. Because transgene expression is not restricted to the pretectum, we used a transient expression strategy to label individual cells by microinjection of a *UAS:CoChR-tdTomato* DNA construct into one-cell stage *KalTA4u508;elavl3:H2B-GCaMP6s* embryos. This allowed us to specifically examine the morphology and function of individual *KalTA4u508*-expressing pretectal neurons. High-contrast membrane labelling by CoChR-tdTomato facilitated morphological reconstruction (at 6–7 dpf) and single-cell tracings were registered to the brain atlas using the H2B-GCaMP6s channel (*Figure 3F, H*).

We identified three morphological classes of pretectal neuron labelled by *KalTA4u508*. One class projects to the ipsilateral optic tectum (*Figure 3F*), elaborating axon terminal arbours preferentially in the most anterior-ventral aspect of OT (N = 4 cells from four fish; *Figure 3G*). The second class makes descending projections to the midbrain and hindbrain tegmentum. Axons decussate near the

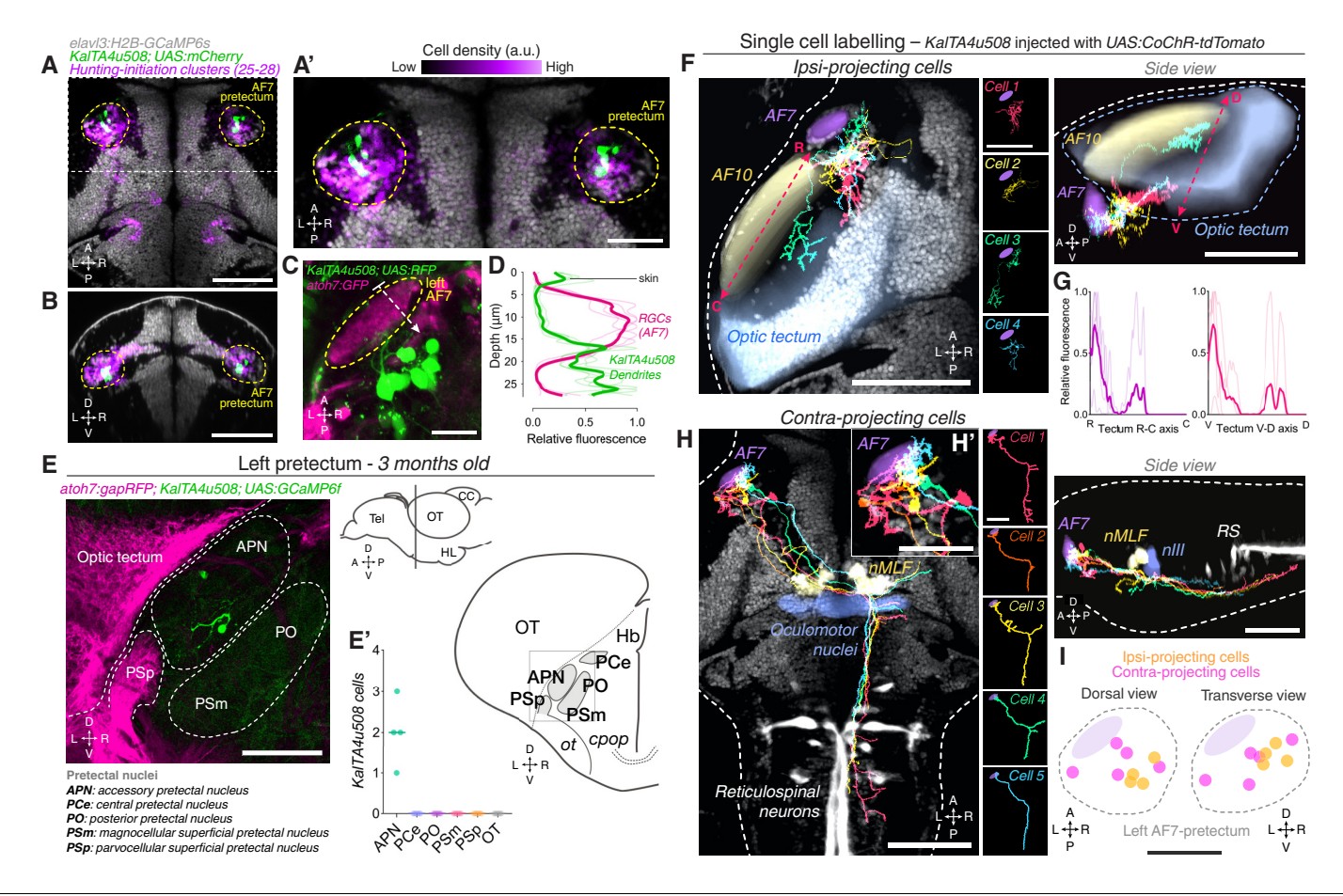

**Figure 3.** AF7-pretectal neurons with distinct projection patterns labelled by *KalTA4u508*. (**A**) Dorsal view of *KalTA4u508;UAS:mCherry* expression at 6 dpf (green) registered to the *elavl3:H2B-GCaMP6s* reference brain (grey). Neurons of all four hunting-initiation clusters combined are shown in purple, colour-coded according to local cell density (clusters 25–28, N = 6,630 cells from eight fish). AF7-pretectum is indicated in yellow and the region is enlarged in (**A'**). (**B**) Ventro-dorsal cross-section of data in (**A**). (**C**) Left AF7-pretectum in a 6 dpf *KalTA4u508;UAS:RFP;atoh7:GFP* larva (dorsal view, maximum-intensity projections, 10 planes, 10 μm depth). (**D**) Dendritic stratification of *KalTA4u508* neurons (green) relative to RGC axons (magenta) in AF7. Y-axis indicates distance from the skin in μm (dashed white arrow in C). Mean and individual stratification patterns are reported (N = 4 fish). (**E**) *KalTA4u508* neurons in pretectum of a 3 month-old *KalTA4u508;UAS:GCaMP6f;atoh7:gapRFP* fish. Pretectal and tectal regions in the left hemisphere are shown. Schematic indicates location of micrograph and pretectal nuclei (transverse plane). Number of *KalTA4u508* cells in each pretectal nucleus are reported in (**E'**) (N = 4 fish). APN, accessory pretectal nucleus; CC, cerebellar corpus; *cpop*, postoptic commissure; Hb, habenula; HL, hypothalamic lobe; OT, optic tectum; *ot*, optic tract; PCe, central pretectal nucleus; PO, posterior pretectal nucleus; PSm, magnocellular superficial pretectal nucleus, PSp, parvocellular superficial pretectal nucleus; Tel, telencephalon. (**F**) Tracings of individual *KalTA4u508* projection neurons that innervate the ipsilateral tectum ('ipsi-projecting' cells, N = 4 cells from four fish). Data is from 6 to 7 dpf larvae and is registered to the *elavl3:H2B-GCaMP6s* reference brain (grey). Selected anatomical regions from the ZBB brain atlas are overlaid. To enable morphological comparisons, all traced neurons are shown in the left hemisphere. (**G**) Fluorescence profiles of neurites of ipsi-projecting *KalTA4u508* cells along the rostro-caudal (R-C, left) and ventral-dorsal (V-D, right) axes of the optic tectum (dashed red arrows in **F**). Mean and individual profiles are reported (N = 4 cells). (**H**) Tracings of *KalTA4u508* projection neurons innervating the contralateral hindbrain ('contra-projecting' cells, N = 5 cells from five fish). Dendritic arbours adjacent to AF7 are enlarged in inset (**H'**). nMLF, nucleus of the medial longitudinal fasciculus; RS, reticulospinal system. (**I**) Soma location of ipsi- and contra-projecting *KalTA4u508* cells in AF7-pretectum. Scale bars, 100 μm, except (**A'**), (**H'**), (**I**), 50 μm, and (**C**), 20 μm. A, anterior; C, caudal; D, dorsal; L, left; P, posterior; R, right (rostral in **G**); V, ventral. See also *Figure 3—figure supplement 1*.

DOI: https://doi.org/10.7554/eLife.48114.011

The following source data and figure supplement are available for figure 3:

**Source data 1.** Source data for *Figure 3*.
DOI: https://doi.org/10.7554/eLife.48114.013

**Figure supplement 1.** *KalTA4u508* neurons innervating cerebellum, and PA-GFP projection mapping from AF7-pretectum.
DOI: https://doi.org/10.7554/eLife.48114.012

nucleus of the medial longitudinal fasciculus (nMLF) and the oculomotor nucleus (nIII) before extending caudally into the contralateral ventral hindbrain. Axon collaterals could be observed bilaterally in the vicinity of nIII/nMLF and proximal to ventral reticulospinal neurons in the contralateral hindbrain (N = 5 cells from five fish; *Figure 3H*). This class of projection neuron extended dendrites within a neuropil region that includes AF7 (*Figure 3H'*), a feature not observed in the other two classes. The third class was characterised by ipsilateral axonal projections to the medial region of the corpus cerebellum (N = 2 cells from two fish; *Figure 3—figure supplement 1B*). Neurite tracing using photoactivatable GFP confirmed a pretectal projection to the cerebellum (*Figure 3—figure supplement 1D*) as well as to nIII/nMLF and contralateral ventral hindbrain (*Figure 3—figure supplement 1C*). This latter projection pattern is compatible with data on APN efferent projections in adult zebrafish (*Yáñez et al., 2018*). In combination with our adult expression data (*Figure 3E*), we conclude that the subset of *KalTA4u508* pretectal neurons projecting to contralateral hindbrain belong to the larval APN.

## Pretectal *KalTA4u508*-expressing neurons have hunting-initiation activity

Next, we asked whether *KalTA4u508* pretectal neurons are responsive to prey-like stimuli and/or are recruited during hunting initiation. We performed 2-photon calcium imaging in *KalTA4u508;UAS: GCaMP6f* or *KalTA4u508;UAS:GCaMP7f* transgenic larvae during the virtual hunting assay (6–7 dpf, N = 30 fish). Notably, *KalTA4u508* pretectal neurons exhibited negligible activity in response to visual stimuli (*Figure 4D* and *Figure 4—figure supplement 1*). Visuomotor vectors were generated for individual cells allowing ~51% to be assigned cluster identities based on the functional archetypes established previously using pan-neuronal imaging (correlation threshold = 0.7, N = 188 out of 369 cells).

Of the *KalTA4u508* pretectal neurons assigned functional identities, 28% were associated with hunting-initiation clusters (clusters 25–28; 52/188 cells; *Figure 4A,B*). The remaining neurons were assigned to either conjugate eye movement clusters (57%) or tail movement clusters (15%) and, in line with the absence of visual sensory responses, no cells were assigned to prey-responsive clusters. Of the hunting-initiation neurons, most *KalTA4u508* cells were associated with functional clusters 26 and 28, which show direction-selective activity (*Figure 4B,E*) and neurons in these two clusters were predominantly located contralateral to the direction of preferred tail movement (73% and 80% contralateral for cluster identities 26 and 28, respectively). The absence of visual activity in *KalTA4u508* neurons is in contrast to the small responses seen previously in hunting-initiation clusters (*Figure 2C*), suggesting some functional heterogeneity in clusters derived from pan-neuronal imaging. *KalTA4u508* cells assigned to hunting-initiation clusters had higher HIx scores than those assigned to other clusters (*Figure 4C*) supporting the hunting-response specificity of their activity.

In summary, *KalTA4u508* provides genetic access to a subset of AF7-pretectal neurons that are selectively active during initiation of hunting behaviour.

## Optogenetic activation of single *KalTA4u508*-expressing pretectal neurons induces hunting

To test whether *KalTA4u508* pretectal cells are capable of inducing predatory behaviour, we optogenetically stimulated single neurons and used high-speed video tracking to monitor free-swimming behaviour (*Figure 5A*). To do this, we used the same larvae described above in which single *KalTA4u508* pretectal cells expressed the optogenetic actuator CoChR-tdTomato (*Figure 5B*) (*Klapoetke et al., 2014*). Experiments consisted of repeated trials, each of 8 s duration, in which larvae (6–7 dpf, N = 70) were exposed to 7 s blue light stimulation (470 nm, 0.44 mW/mm$^2$), interleaved with trials with no stimulation.

Strikingly, we found that optogenetic stimulation of individual *KalTA4u508* pretectal neurons could induce sustained, hunting-like behavioural routines. As in naturalistic hunting, optogenetically induced hunting routines comprised a sequence of swim bouts that coincided with a sustained period of elevated ocular vergence (*Figure 5C–E*). These sequences were initiated by a convergent saccade accompanied by a lateralised swim bout (*Figure 5C*) and in several cases terminated with a bout resembling a capture swim and paired with jaw opening (*Figure 5—figure supplement 1C,D*). The fraction of *KalTA4u508* pretectal neurons that induced hunting (32%, 23 out of 70 cells) was

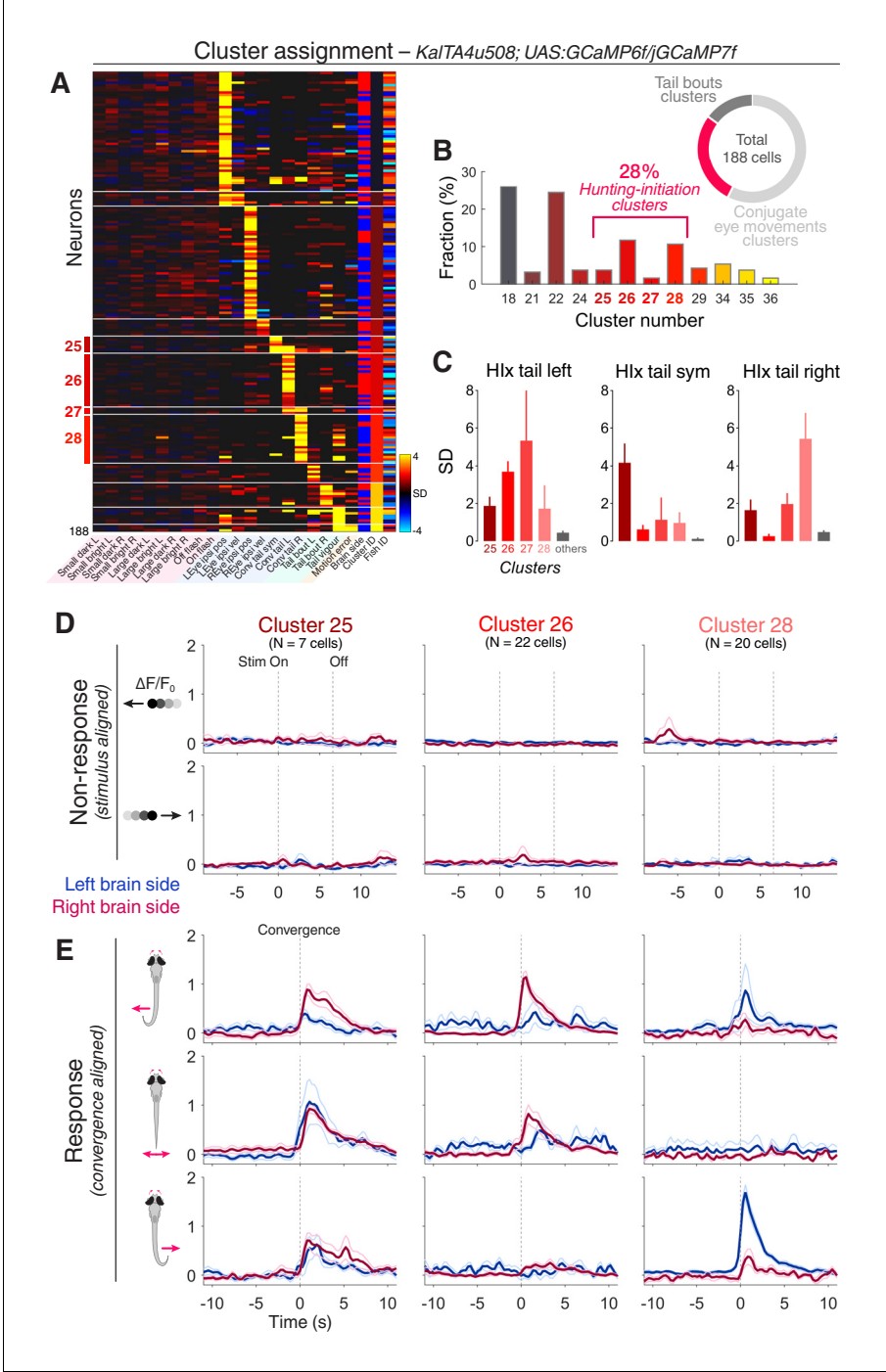

**Figure 4.** *KalTA4u508* pretectal neurons are active during hunting initiation. (**A**) VMVs of *KalTA4u508* neurons with assigned cluster identities (N = 188 neurons from 30 fish). Cell location (blue for left hemisphere, red for right) is reported by the 'Brain side' column. L, left; R, right; Sym, symmetric. (**B**) Fraction of assigned *KalTA4u508* neurons in each cluster. (**C**) Hunting Index (HIx) scores for *KalTA4u508* neurons in different clusters (mean + SEM). (**D**) Visual stimulus-aligned responses of *KalTA4u508* neurons during non-response trials (mean ± SEM). Traces are colour-coded according to anatomical laterality (blue for left hemisphere, red for right). Stim, stimulus. (**E**) Eye convergence-aligned neuronal responses. Activity during both spontaneous and visually evoked convergent saccades was used. See also *Figure 4—figure supplement 1*.

DOI: https://doi.org/10.7554/eLife.48114.014

The following source data and figure supplement are available for figure 4:

**Source data 1.** Source data for *Figure 4*.

*Figure 4 continued on next page*

*Figure 4 continued*

DOI: https://doi.org/10.7554/eLife.48114.016

**Figure supplement 1.** Visual responses and activity during spontaneous convergences of *KalTA4u508* pretectal neurons.

DOI: https://doi.org/10.7554/eLife.48114.015

similar to the proportion that were assigned to hunting-initiation clusters (28%; *Figure 4B*). Hunting-like responses were entirely dependent on blue light stimulation. For responsive fish, we observed 18.1% median response probability in LED-On trials vs. 0% in LED-Off trials (p<0.0001, N = 23 fish; *Figure 5F*). Furthermore, control experiments demonstrated that opsin-negative animals do not produce hunting behaviours in response to blue light stimulation alone (0/16 animals; *Figure 5—figure supplement 1I*). By examining single-cell morphology, we found that the *KalTA4u508* cells that could evoke hunting behaviour belonged to the projection classes that innervated the ipsilateral optic tectum (9/23 cells, hereafter abbreviated 'ipsi-projecting') or that belong to the presumptive APN and connect to the contralateral tegmentum (14/23 cells, 'contra-projecting'; *Figure 5G,H*).

How closely do the hunting-like routines induced by optogenetic stimulation of *KalTA4u508* pretectal neurons compare to hunting behaviour? To address this question, we compared a variety of oculomotor and locomotor kinematics between optogenetically induced hunting routines versus hunting of live *Paramecia*. The duration of hunting sequences, defined by the period of elevated ocular vergence, as well as vergence angles, were equivalent between natural and optogenetically evoked behaviour (*Figure 5I,J*). In 100% of optogenetically-evoked hunting routines, a lateralised swim bout immediately followed saccadic convergence with short latency, comparable to natural hunting (*Figure 5K*). We analysed several kinematic features of these initiating bouts, which resemble 'J-turns', focussing on features that distinguish hunting swims from spontaneous exploratory behaviour. This revealed a high degree of kinematic similarity between natural hunting bouts and optogenetically induced hunting bouts. In both cases, bouts comprised a highly lateralised sequence of half-beats ('bout asymmetry'; *Figure 5L*, see Materials and methods for details), a large fraction of curvature was localised to the distal segments of the tail (*Figure 5M*) and bouts displayed low tail beat frequency (*Figure 5N*). Notably, all such parameters were significantly different as compared to spontaneous swims. Optogenetically induced hunting bouts showed a reduction in average vigour and theta-1 angles (maximum tail angle during the first half beat) compared to bouts during *Paramecia* hunting (*Figure 5O,P*). However, the latter displayed a bimodal distribution of theta-1 and optogenetically induced hunting bouts overlapped with the lower amplitude component of this distribution (*Figure 5P*, bottom). In sum, our data indicate that optogenetic stimulation of single *KalTA4u508* pretectal neurons can induce naturalistic hunting-like behaviour.

Next, we compared hunting routines induced by optogenetic activation of ipsi- versus contra-projecting *KalTA4u508* neurons. We did not observe differences in response latency (*Figure 5Q*), duration of evoked hunting sequences or number of swim bouts they contained (*Figure 5—figure supplement 1A,B*) or oculomotor kinematics (*Figure 5—figure supplement 1E–H*). A similar fraction (~40%) of both types of pretectal neuron could drive hunting sequences that terminated with capture swims (*Figure 5—figure supplement 1C,D*). However, the laterality of evoked hunting responses differed between the two projection classes (*Figure 5R,S*): Stimulation of ipsi-projecting *KalTA4u508* pretectal neurons most frequently induced hunting in which the first swim bout was oriented in the ipsilateral direction (*i.e.* a left pretectal neuron evoked leftward turning). By contrast, contra-projecting neurons most frequently induced contralaterally directed hunting responses.

To directly establish whether the *KalTA4u508* pretectal neurons that can induce predatory behaviour are the same cells that are recruited during visually evoked hunting, we combined optogenetic stimulation and functional calcium imaging of single neurons. To achieve this, we first established that optogenetic stimulation of a given *KalTA4u508* pretectal neuron could induce hunting and then tethered the larva in agarose and performed calcium imaging of H2B-GCaMP6s, expressed in the nucleus of the same neuron, while the animal engaged in the virtual hunting assay. Visuomotor fingerprinting and cluster assignment of these neurons showed that they all belonged to hunting-initiation clusters (clusters 25–27) and had high HIx scores (N = 6 cells from six fish; *Figure 5T*).

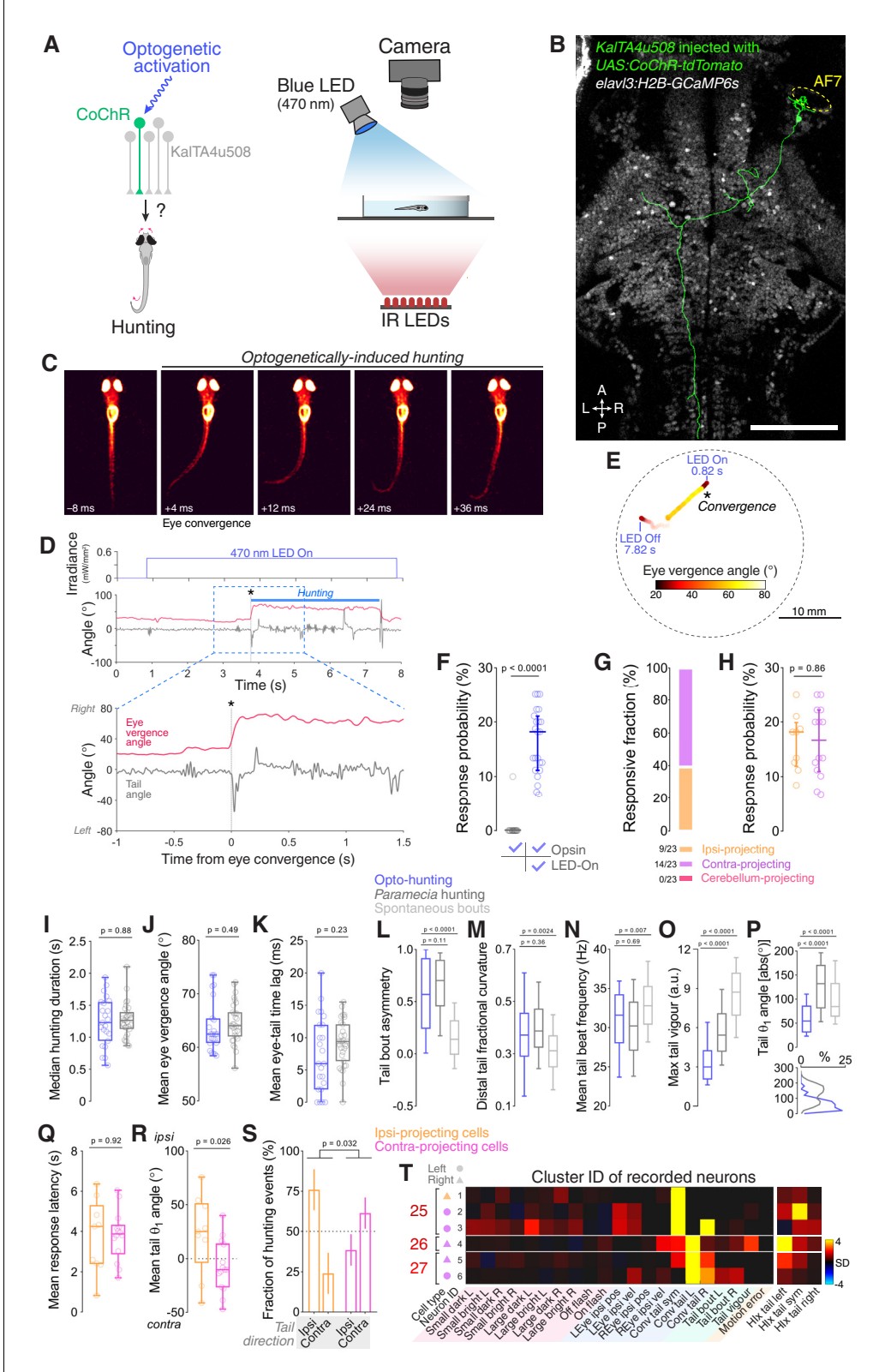

**Figure 5.** Optogenetic stimulation of single *KalTA4u508* pretectal neurons induces hunting. (**A**) Optogenetic stimulation of single neurons paired with behavioural tracking. (**B**) A single *KalTA4u508* neuron in a 7 dpf *KalTA4u508;elavl3:H2B-GcaMP6s* larva that was injected at the one-cell stage with *UAS: CoChR-tdTomato* DNA. This 'contra-projecting' neuron is 'Cell 3' in *Figure 3H*. A, anterior; L, left; P, posterior; R, right. Scale bar, 100 μm. (**C**) Example frames from an optogenetically induced hunting event. Labels indicate time relative to saccadic eye convergence/hunting initiation. (**D**) Tail angle (grey)

*Figure 5 continued*

and ocular vergence angle (red) during an optogenetically induced hunting event ('ipsi-projecting' cell located in the left AF7-pretectum; this neuron is 'Cell 4' in *Figure 3F*; see also *Video 2*). Asterisk indicates time of convergent saccade. (E) Larval location colour-coded by vergence angle during the example hunting event in (D). (F) Hunting response probability for LED-On versus non-stimulation trials for larvae that performed at least one eye convergence during optogenetic stimulation (N = 23 fish). (G) Morphological identity of *KalTA4u508* neurons that elicited hunting upon optogenetic stimulation. Numbers of responsive larvae are reported at the bottom (N = 23 fish). (H) Hunting response probability upon optogenetic stimulation of ipsi-projecting (orange, N = 9 cells) and contra-projecting neurons (magenta, N = 14). (I–P) Comparison of behavioural kinematics between optogenetically induced hunting events (blue, N = 23 fish) and *Paramecia* hunting (dark grey, N = 31). Tail kinematics for non-hunting swim bouts were recorded from the same larvae that were monitored during *Paramecia* hunting (light grey). In (L–P), data from all bouts are plotted, whereas in (I–K) the median or mean for each larva is reported. (Q–S) Behavioural kinematics of hunting events induced by stimulation of ipsi-projecting *KalTA4u508* neurons (orange, N = 9 cells) or contra-projecting neurons (magenta, N = 14). (T) VMVs and cluster identities of *KalTA4u508* neurons that induced hunting upon optogenetic stimulation and subsequently underwent calcium imaging (N = 6 cells from six fish). Symbols on the left indicate projection cell class and left/right location. HIx scores are shown on right. See also *Figure 5—figure supplements 1,2* and *Video 2*.

DOI: https://doi.org/10.7554/eLife.48114.017

The following source data and figure supplements are available for figure 5:

**Source data 1.** Source data for *Figure 5*.
DOI: https://doi.org/10.7554/eLife.48114.020
**Figure supplement 1.** Behavioural kinematics of optogenetically induced hunting and responses at increased irradiance.
DOI: https://doi.org/10.7554/eLife.48114.018
**Figure supplement 2.** Optogenetic stimulation of the *KalTA4u508* pretectal population induces hunting with short latency.
DOI: https://doi.org/10.7554/eLife.48114.019

## Optogenetic activation of multiple *KalTA4u508*-expressing pretectal neurons induces short-latency hunting responses

Optogenetic stimulation of single *KalTA4u508* pretectal neurons induced hunting routines with relatively low response probability (~20%) and long latency (~4 s). We investigated if response rates might be increased by stimulating a larger number of pretectal neurons or increasing light intensity.

First, we optogenetically stimulated single *KalTA4u508* pretectal neurons at a range of irradiance levels (0.44–2.55 mW/mm$^2$, 459 nm). To test whether the bright blue stimulation light might visually interfere with induction of hunting, we also tested retinally blind '*lakritz*' larvae, homozygous for the atoh7$^{th241}$ mutation, in which no RGCs are generated (*Kay et al., 2001*). In one of the two blind larvae in which we could optogenetically evoke hunting, response rate was positively modulated by irradiance but mean response latency did not fall below 2 s (*Figure 5—figure supplement 1I,J*). Although preliminary, this suggests that single pretectal cells may not be able to recruit downstream motor pattern generator circuits to execute hunting programmes with short latency.

Optogenetic activation of multiple *Kal-TA4u508* pretectal neurons reliably evoked hunting behaviour at short latency. We showed this by using patterned illumination to focally activate the AF7-pretectal region in tethered *Kal-TA4u508;UAS:CoChR-tdTomato* transgenic larvae (7 dpf, 4 s LED-On periods, 470 nm, 21.1 mW/mm$^2$; *Figure 5—figure supplement 2A,B*). We unilaterally stimulated the population of transgenically labelled neurons in the AF7-pretectal region and observed convergent saccades paired with lateralised tail movements with high response probability (55.43 ± 35.26%, mean ± SD, N = 7 larvae) and short latency (82.87 ± 5.46 ms, mean ± SD; *Figure 5—figure supplement 2C–E*). Every convergent saccade was associated with tail movement, which tended to be directed contralateral to the stimulated pretectum (*Figure 5—figure supplement*

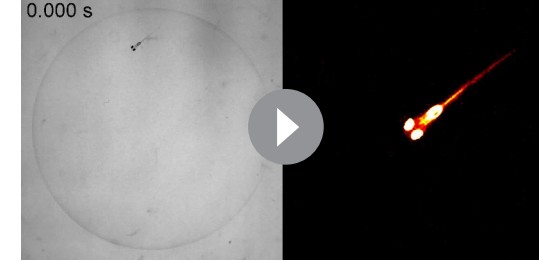

**Video 2.** Hunting behaviour evoked by optogenetic stimulation of a single *KalTA4u508* pretectal neuron. Data from a 6 dpf larva in which a single ipsi-projecting *KalTA4u508* neuron, located in the left AF7-pretectum (Cell 4 in *Figure 3F*), expressed CoChR. Tracking data is reported in *Figure 5D,E*. The video was acquired at 250 frames per second and plays at 0.4 times the original speed. The raw movie is showed on the left, and a background-subtracted inset centred on the larva is showed on the right. Related to *Figure 5*.
DOI: https://doi.org/10.7554/eLife.48114.021

*2F*) similar to activation of single contra-projecting *KalTA4u508* cells. No hunting-like behaviour was induced by stimulation of an off-target region or in opsin-negative animals (N = 7 larvae; *Figure 5—figure supplement 2D*).

In summary, *KalTA4u508* labels a specific group of pretectal neurons that are recruited during hunting initiation and which are capable of inducing a naturalistic predatory motor programme in the complete absence of prey.

## Ablation of *KalTA4u508*-expressing pretectal neurons impairs hunting

To what extent are *KalTA4u508* pretectal neurons required for hunting? To address this question, we assessed hunting performance in freely swimming larvae provided with *Paramecia*, both before and after laser-ablation of *KalTA4u508* pretectal neurons (*Figure 6A–C*). To enable evaluation of the specificity of behavioural phenotypes, we also presented looming stimuli and drifting gratings to test visually evoked escape and optomotor response (OMR), respectively. Ablations were performed at 6 dpf in *KalTA4u508;UAS:mCherry;elavl3:itTA;Ptet:ChR2-YFP* larvae and their efficacy was confirmed by reimaging the pretectum the following day. We estimated that ~90% of the fluorescently labelled *KalTA4u508* pretectal population was typically ablated in both brain hemispheres (*Figure 6D,E*). Importantly, RGC axons in AF7 were not damaged (*Figure 6—figure supplement 1A*). We also evaluated two sets of control larvae: 'Non-ablated', which underwent the same manipulations other than laser ablation, and 'sham ablated' in which an equivalent number of cells were laser-ablated in the thalamus, medially adjacent to AF7-pretectum (*Figure 6—figure supplement 1B–D*).

Analysis of prey consumption revealed that ablation of *KalTA4u508* pretectal neurons resulted in decreased hunting performance (*Figure 6F*). Further analysis revealed that this reduction in prey-capture was associated with a reduced rate of hunting initiation (*Figure 6G*) as well as a reduction in the duration of hunting routines (*Figure 6H*). By contrast, we did not observe changes in average swim speeds, loom-evoked escapes or OMR performance (*Figure 6I–L*) and control larvae did not show changes in any of the tested behaviours (*Figure 6—figure supplement 1E–L*).

Together, these results indicate that *KalTA4u508* pretectal neurons are specifically required for normal initiation and maintenance of predatory behaviour.

## Optogenetic stimulation of optic tectum evokes hunting-like behaviour

Optogenetic stimulation of the anterior-ventral optic tectum (avOT) in *elavl3:itTA;Ptet:ChR2-YFP* transgenic larvae has been previously reported to induce convergent saccades and J-turns (*Fajardo et al., 2013*). To confirm this and explore a potential interaction between avOT and *KalTA4u508* pretectal neurons, we first examined ChR2-YFP expression in 6 dpf *elavl3:itTA;Ptet:ChR2-YFP* larvae. We found that the opsin is highly expressed in avOT, as previously reported, as well as AF7 (*Figure 7B* and *Figure 7—figure supplement 1A*), but is absent from *KalTA4u508* pretectal neurons (*Figure 7D*). Expression in AF7 raised the possibility that stimulation of the axon terminals of the prey-responsive RGCs that innervate this AF (*Semmelhack et al., 2014*) might contribute to the hunting behaviour observed in this transgenic line. We therefore crossed *elavl3:itTA;Ptet:ChR2-YFP* to the atoh7$^{th241}$ mutant to eliminate RGCs (*Figure 7C*). In blind (atoh7$^{-/-}$) transgenic animals, ChR2 expression was reduced in retinal arborisation fields, as expected, but retained in avOT (*Figure 7—figure supplement 1B*). We subsequently examined whether optogenetic stimulation of atoh7$^{-/-}$ transgenic animals was able to induce hunting-like behaviour (*Figure 7A*).

Strikingly, we found that blind *elavl3:itTA;Ptet:ChR2-YFP* transgenics were more likely to display optogenetically induced hunting than their sighted (atoh7$^{+/+}$ or atoh7$^{+/-}$) siblings (29/30 responsive blind larvae vs. 13/30 sighted larvae; *Figure 7E–H* and *Video 3*). In addition, responsive atoh7$^{-/-}$ transgenics showed an increased probability of optogenetically induced hunting events, longer hunting routine durations and a substantial reduction in response latency (*Figure 7I–K*). When hunting was evoked, blind and sighted larvae showed comparable eye and tail kinematics (*Figure 7L* and *Figure 7—figure supplement 1C–F*).

These data are compatible with the conclusion of *Fajardo et al. (2013)*, namely that optogenetic stimulation of avOT elicits hunting in *elavl3:itTA;Ptet:ChR2-YFP* larvae and indicate that RGC stimulation (either visually with bright blue light, or optogenetically) interferes with induction of hunting responses.

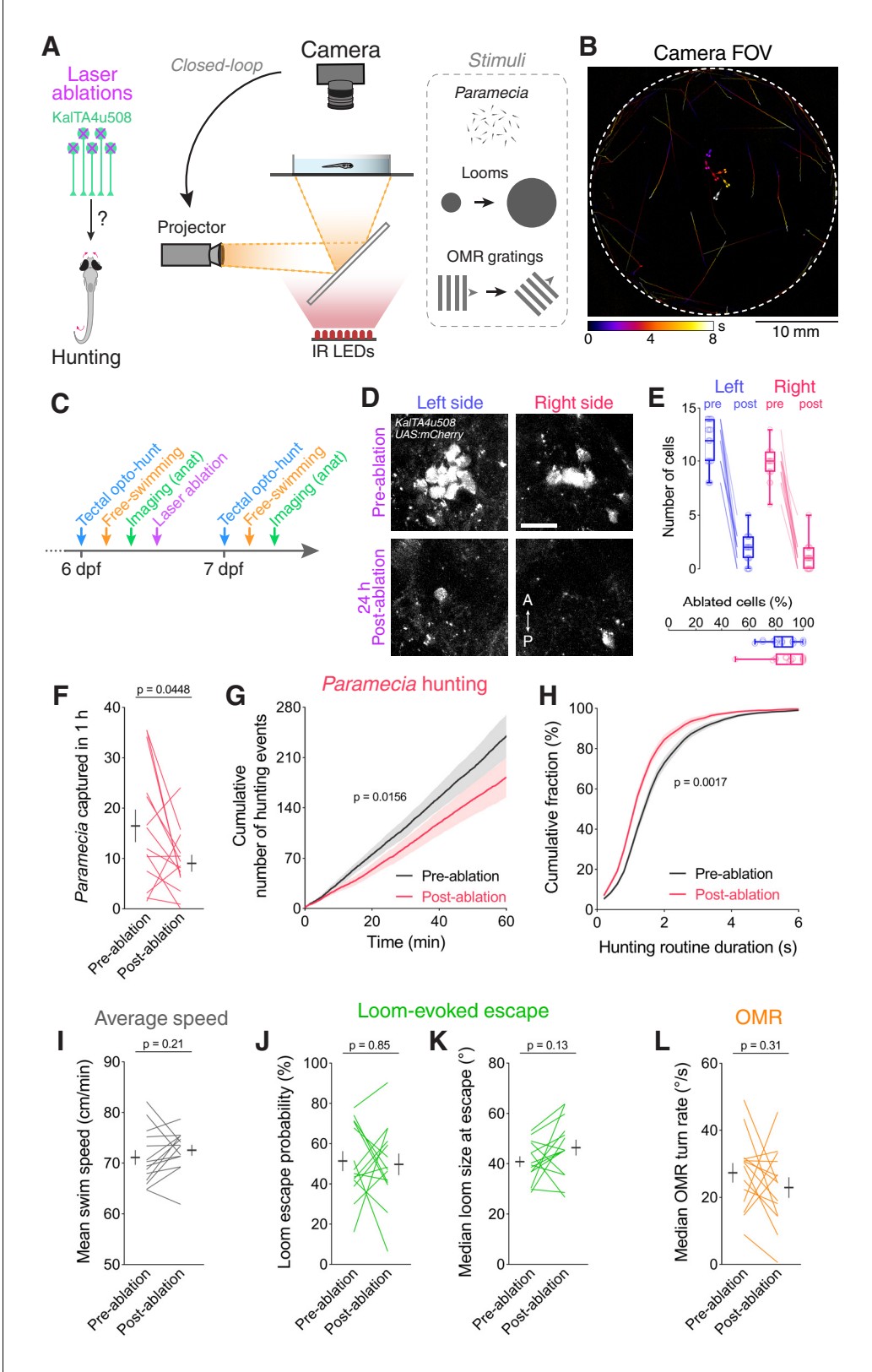

**Figure 6.** Ablation of *KalTA4u508* pretectal neurons impairs hunting. (**A**) Laser ablation of *KalTA4u508* pretectal neurons and assessment of visuomotor behaviours. (**B**) Time-projection of larval behaviour (duration 8 s) showing trajectories of *Paramecia* and larval zebrafish swimming in the arena. (**C**) Time course of behavioural tests, ablation

*Figure 6 continued on next page*

*Figure 6 continued*

and brain imaging. (**D**) Pretectal neurons before (top, 6 dpf) and 24 hr after (bottom, 7 dpf) bilateral laser-ablations in a *KalTA4u508;UAS:mCherry;elavl3:itTA;Ptet:ChR2-YFP* larva. Images show maximum-intensity projections (red channel, 10 planes, 10 µm depth). A, anterior; P, posterior. Scale bar, 20 µm. (**E**) Quantification of cell ablation in left (blue) and right (red) AF7-pretectum (N = 14 fish). (**F–H**) Assessment of hunting performance before and after bilateral ablation of *KalTA4u508* neurons (N = 14 fish). Mean ± SEM is reported for each condition. (**I–L**) Average swim speed, loom-evoked escape and OMR behaviour, before and after ablations. See also ***Figure 6—figure supplement 1***.

DOI: https://doi.org/10.7554/eLife.48114.022

The following source data and figure supplement are available for figure 6:

**Source data 1.** Source data for *Figure 6*.
DOI: https://doi.org/10.7554/eLife.48114.024

**Figure supplement 1.** Assessment of visuomotor behaviours in control larvae.
DOI: https://doi.org/10.7554/eLife.48114.023

## *KalTA4u508*-expressing pretectal neurons are required for tectally evoked hunting behaviour

Does hunting behaviour evoked by tectal activation require AF7-pretectal circuits? To investigate this, we tested whether hunting could be evoked by optogenetic stimulation of avOT in larvae in which *KalTA4u508* pretectal cells were ablated (***Figure 8A***). Following laser-ablation of *KalTA4u508* neurons in the same *elavl3:itTA;Ptet:ChR2-YFP;KalTA4u508;UAS:mCherry* larvae tested above, we observed that the probability of optogenetically induced hunting was substantially reduced (***Figure 8B***). By contrast, non-ablated control larvae showed no change in response probability between 6 and 7 dpf (***Figure 6—figure supplement 1M***).

These data indicate that *KalTA4u508* pretectal neurons are required for release of predatory behaviour by circuits in the anterior optic tectum.

## Discussion

In this study, we combined multi-photon calcium imaging, optogenetic stimulation and laser-ablations to identify a population of pretectal neurons that controls hunting behaviour. Calcium imaging during naturalistic behaviour revealed that *KalTA4u508*-expressing pretectal neurons are recruited when larval zebrafish initiate hunting. Optogenetic activation of single *KalTA4u508* pretectal cells could release predatory behaviour in the absence of prey, and ablation of these cells impaired both natural hunting as well as hunting-like behaviour evoked by avOT stimulation. Based on functional and anatomical data, we propose that *KalTA4u508* pretectal cells function as a command system linking visual perception of prey-like stimuli to activation of tegmental pattern generating circuits that coordinate specialised hunting motor programmes.

### Do AF7-pretectal neurons comprise a command system controlling predation?

We suggest that *KalTA4u508* pretectal neurons satisfy the criteria for a command system for induction of predatory behaviour. A command neuron has been defined as 'a neuron that is both necessary and sufficient for the initiation of a given behaviour' (***Kupfermann and Weiss, 1978***). Although a small number of striking examples of such cells have been identified in invertebrate models (***Frost and Katz, 1996***; ***Flood et al., 2013***), it is recognised that the 'necessity' criterion is unlikely to be fulfilled in larger nervous systems where individual neurons display functional redundancy (***Yoshihara and Yoshihara, 2018***). Thus, command systems (sometimes referred to as decision neurons, command-like neurons or higher-order neurons) have been defined as interneurons that are active in association with a well-defined behaviour and whose activity can induce that behaviour, but without the strict necessity requirement (***Jing, 2009***; ***Yoshihara and Yoshihara, 2018***).

To what extent do *KalTA4u508* pretectal neurons satisfy these criteria? First, we showed that these cells are recruited during naturalistic hunting. By comparing neural activity in response versus non-response trials we were able to disambiguate 'sensory' activity, evoked by prey-like visual cues, from activity specifically associated with *execution* of hunting behaviour. We discovered that

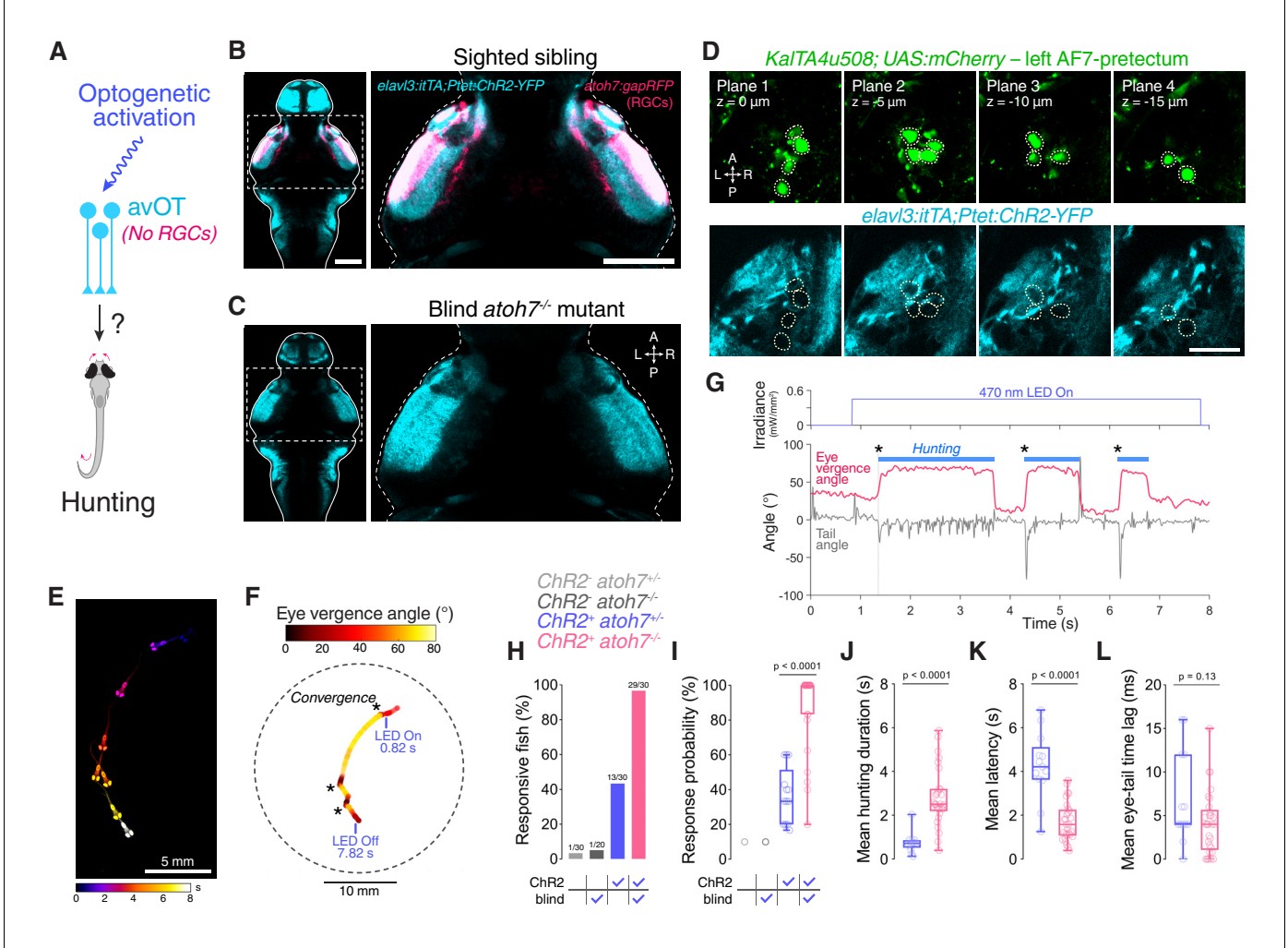

**Figure 7.** Optogenetic stimulation of avOT induces hunting in the absence of RGCs. (A) Optogenetic stimulation of anterior-ventral optic tectum (avOT). (B–C) Dorsal view of ChR2-YFP expression in sighted (B) and blind *atoh7^-/-^* (C) 6 dpf *elavl3:itTA;Ptet:ChR2-YFP;atoh7:gapRFP* larvae registered to the ZBB brain atlas. Axonal projections of RGCs labelled by the *atoh7:gapRFP* transgene are displayed in magenta and are absent in *atoh7^-/-^* larvae. Images derive from median datasets across multiple immunostained brains (N = 6 sibling and N = 7 blind fish) and show maximum-intensity projections through focal planes encompassing AF7-pretectal regions. (D) ChR2 expression relative to *KalTA4u508* neurons in a 6 dpf *KalTA4u508;UAS:mCherry; elavl3:itTA;Ptet:ChR2-YFP* larva. Images are single focal planes obtained from the left AF7-pretectum (plane one is dorsal relative to the other z-planes). *KalTA4u508* neurons are not labelled with ChR2-YFP. (E) Time sequence composite image showing selected frames from an example optogenetically induced hunting sequence in a blind *atoh7^-/-^* 6 dpf *elavl3:itTA;Ptet:ChR2-YFP;atoh7:gapRFP* larva (see also *Video 3*). (F) Vergence angle overlaid onto larval location during the optogenetically induced hunting sequence from (E). Asterisks indicate time of convergent saccades. (G) Tail angle (grey) and ocular vergence angle (red) during the optogenetically induced hunting sequence shown in (E) and (F). Asterisks indicate time of convergent saccades. (H) Fraction of larvae that performed eye convergences during optogenetic stimulations. Larvae were either sighted (*atoh7^+/+^* or *atoh7^+/-^*) or blind (*atoh7^-/-^*) and either opsin-positive (*elavl3:itTA;Ptet:ChR2-YFP;atoh7:gapRFP*) or opsin-negative (*atoh7:gapRFP*). Numbers of responsive larvae are reported above bars. (I) Response probability of larvae that performed eye convergence in at least one optogenetic stimulation trial. (J–L) Behavioural kinematics for optogenetically induced hunting events elicited in sighted (blue, N = 13 fish) and blind *atoh7^-/-^* (pink, N = 29) *elavl3:itTA;Ptet:ChR2-YFP; atoh7:gapRFP* larvae. Scale bars, 100 μm, except in (D), 30 μm. A, anterior; L, left; P, posterior; R, right. See also *Figure 7—figure supplement 1* and *Video 3*.

DOI: https://doi.org/10.7554/eLife.48114.025

The following source data and figure supplement are available for figure 7:

**Source data 1.** Source data for *Figure 7*.
DOI: https://doi.org/10.7554/eLife.48114.027

**Figure supplement 1.** ChR2 expression in *elavl3:itTA;Ptet:ChR2-YFP;atoh7:gapRFP* larvae and additional behavioural kinematics of optogenetically induced hunting.
DOI: https://doi.org/10.7554/eLife.48114.026

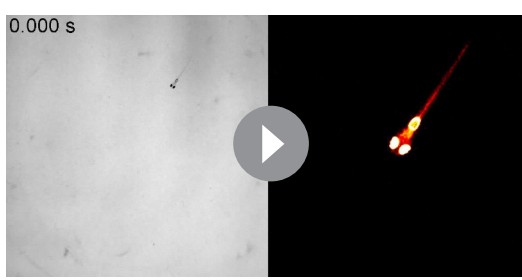

**Video 3.** Hunting behaviour evoked by optogenetic stimulation of the anterior-ventral optic tectum in a blind larva. Optogenetically induced hunting behaviour in a blind *atoh7⁻/⁻* 6 dpf *elavl3:itTA;Ptet:ChR2-YFP; atoh7:gapRFP* larva. Tracking data is reported in *Figure 7E–G*. The video was acquired at 250 frames per second and plays at 0.4 times the original speed. The raw movie is showed on the left, and a background-subtracted inset centred on the larva is showed on the right. Related to *Figure 7*.
DOI: https://doi.org/10.7554/eLife.48114.028

*KalTA4u508* pretectal neurons, located close to AF7, display no detectable activity in response to prey-like stimuli but are reliably activated when larvae release convergent saccades at the commencement of hunting. This activity signature is compatible with a premotor role in directing predatory actions, as opposed to visual encoding of prey.

Strikingly, optogenetic stimulation of single *KalTA4u508* neurons could evoke sustained hunting routines. As such, this is one of the first examples of a behavioural action sequence induced by activation of a single neuron in a vertebrate. Crucially, this induction occurred in the absence of any prey-like stimulus, supporting a role for these neurons in release of predatory behaviour downstream of perceptual recognition of prey. Optogenetic stimulation of single cells allowed us to test behaviour in freely swimming larvae and interrogate the roles of two distinct classes of AF7-pretectal cell. However, hunting was induced with surprisingly long average latency (~4 s) and in only ~20% of trials. In contrast, optogenetic stimulation directed at the entire transgenically labelled AF7-pretectal population reliably induced behaviour with very short latency,~80 ms. These findings are compatible with the *KalTA4u508* population comprising a command *system* controlling hunting and suggest that during natural behaviour, presumed downstream pattern generating circuits are unlikely to be brought to threshold by the action of single pretectal neurons.

Ablations targeting *KalTA4u508* pretectal neurons impaired, but did not eliminate, hunting. One contributing factor is likely to be that ablations were incomplete. We observed that 10% of labelled *KalTA4u508* neurons survived ablation and due to variegation of transgene expression (*Akitake et al., 2011*), this is probably a lower bound on the size of the surviving pretectal population. Notably, the observation that hunting can be evoked by stimulation of only a single neuron suggests few surviving cells could in principle suffice to release predatory behaviour. Although hunting evoked by stimulation of anterior-ventral OT was strongly diminished in *KalTA4u508*-ablated animals, we do not rule out the possibility that there might be redundant, distributed circuitry involved in hunting initiation.

Overall, our data support the conclusion that a discrete population of pretectal neurons comprises a command system controlling predatory hunting.

## Neural circuitry controlling visually guided hunting

How might these pretectal neurons fit within a sensorimotor pathway controlling visually guided hunting (*Figure 8C*)? Current evidence suggests visual recognition of prey is mediated by tectal and/or pretectal circuits. The axon terminals of zebrafish 'projection class 2' retinal ganglion cells in AF7 appear tuned to prey-like stimuli (*Semmelhack et al., 2014*) and a subpopulation of tectal neurons show non-linear mixed selectivity for conjunctions of visual features that best evoke predatory responses (*Bianco and Engert, 2015*). In accordance with these findings, we observed prey-responsive neurons in tectum and AF7-pretectum which were activated by small moving spots regardless of whether or not the animal produced a hunting response. Ablations of either AF7 or tectal neuropil have been shown to substantially impair hunting (*Gahtan et al., 2005*; *Semmelhack et al., 2014*), compatible with an important function for these retinorecipient regions in visual perception of prey.

Command systems are thought to sit at the sensorimotor 'watershed', linking sensory processing to activation of motor hierarchies. Our data support a circuit organisation where downstream of prey detection, the recruitment of *KalTA4u508* pretectal neurons might be the neural event that corresponds to the decision of the animal to initiate hunting. The dendritic arbours of these pretectal neurons lie in immediate apposition to RGC terminals in AF7, suggesting a biological interface for

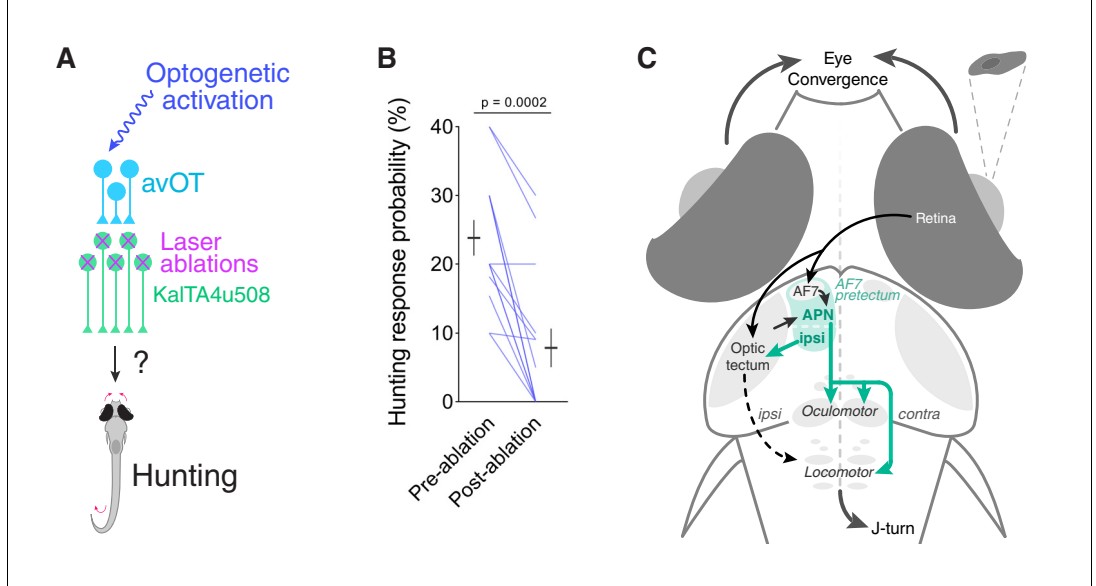

**Figure 8.** *KalTA4u508* pretectal neurons are required for tectally induced hunting. (**A**) Optogenetic stimulation of avOT before and after ablation of *KalTA4u508* pretectal neurons. (**B**) Hunting response probability upon optogenetic stimulation of *KalTA4u508;UAS:mCherry;elavl3:itTA;Ptet:ChR2-YFP* larvae, before and after bilateral ablation of *KalTA4u508* pretectal neurons (N = 14 fish). (**C**) Model of the neural circuit that induces hunting behaviour. Two classes of AF7-pretectal projection neuron are capable of inducing hunting. Contralaterally projecting APN neurons are likely to induce hunting by recruiting activity in oculomotor and locomotor pattern generating circuits in the mid/hindbrain tegmentum. Ipsilaterally projecting AF7-pretectal neurons may recruit ipsilateral tectofugal pathways. Activation of anterior-ventral tectum requires AF7-pretectal neurons to induce hunting and likely operates via the contralaterally projecting APN population because unilateral avOT stimulation produces contralaterally directed responses (*Fajardo et al., 2013*). APN, accessory pretectal nucleus.

DOI: https://doi.org/10.7554/eLife.48114.029

The following source data is available for figure 8:

**Source data 1.** Source data for *Figure 8*.
DOI: https://doi.org/10.7554/eLife.48114.030

transforming visual sensory activity into premotor commands to release behaviour. Photoactivation data also suggests the AF7-pretectal region is closely interconnected with ipsilateral OT (*Figure 3—figure supplement 1E*), providing a further route for visual prey detectors in OT to recruit a hunting response. Consistent with this, our ablation data indicate that induction of hunting by stimulation of anterior-ventral OT (*Fajardo et al., 2013*) requires *KalTA4u508* pretectal neurons.

Two morphological classes of *KalTA4u508* neuron were able to induce predatory behaviour with similar efficacy and motor kinematics, but opposite directional biases. Both morphologies appear similar to neurons previously identified in the AF7 region by *Semmelhack et al. (2014)*. Contralaterally projecting neurons extended axon collaterals around the nMLF and oculomotor nuclei as well as in close apposition to reticulospinal neurons in the hindbrain. This projection pattern, as well as the identification of *KalTA4u508* neurons in adult brain sections, suggests these cells reside in the larval zebrafish APN. Their axonal projections provide a pathway for recruitment of tegmental motor pattern generators to produce the specialised eye and tail movements observed during hunting (*McElligott and O'malley, 2005*; *Bianco et al., 2011*; *Marques et al., 2018*). Optogenetic stimulation of ipsilaterally projecting *KalTA4u508* neurons evoked hunting routines in which the first orienting turn displayed an ipsilateral bias. It is possible that this arises from recruitment of tectal efferent pathways to the ipsilateral tegmentum, which have recently been suggested to mediate prey-directed orienting turns (*Helmbrecht et al., 2018*).

Whilst it is remarkable that stimulation of single pretectal neurons could induce hunting-like behaviour in the absence of prey, natural hunting is a precise orienting behaviour directed towards a (visual) target. Optogenetically induced hunting routines involved swim bouts where turn angle fell within the lower portion of the distribution measured during prey hunting. A probable explanation is that in the presence of prey, appropriate steering signals derive from OT to guide lateralised

orienting turns. Compact tectal assemblies show premotor activity immediately preceding hunting initiation and are anatomically localised in relation to the directionality of hunting responses (*Bianco and Engert, 2015*). Furthermore, focal stimulation of the retinotopic tectal map has long been known to evoke goal-directed orienting responses including predatory manoeuvres (*Herrero et al., 1998*; *Bels et al., 2012*) and suppression of specific populations of neurons in superior colliculus (SC) impairs visual prey detection and accurate pursuit (*Hoy et al., 2019*). We hypothesise that pretectal activity releases predatory behaviour and operates in parallel with prey-directed steering signals, most likely from OT.

Command systems have been identified that evoke both discrete actions (*Korn and Faber, 2005*) as well as entire behavioural programmes (*Flood et al., 2013*). Constant stimulation of pretectal *KalTA4u508* neurons produces extended hunting sequences that commenced with saccadic eye convergence followed by serial execution of multiple swim bouts during which elevated ocular vergence – a hallmark of predatory behaviour – was maintained. In some cases, optogenetically induced hunting sequences even terminated in capture swims associated with jaw movements. This suggests that the function of pretectal activity is to induce the overall hunting programme and that individual component actions (*i.e.* discrete tracking swim bouts) are coordinated by downstream motor pattern generating circuits.

### Induction and modulation of predatory behaviour

In recent years, several studies in rodents have demonstrated that pathways converging on the periaqueductal grey (PAG) can potently modulate predatory behaviour. The central amygdala (CeA) displays activity changes during hunting (*Comoli et al., 2005*) and optogenetic activation of the CeA→ventral PAG pathway motivates prey pursuit (*Han et al., 2017*). Stimulation of the medial preoptic area to vPAG pathway promotes acquisition/handling (grabbing, biting) of objects, including prey (*Park et al., 2018*), and activation of a GABAergic projection from lateral hypothalamus to lateral/ventrolateral PAG motivates attack on prey (*Li et al., 2018*). In these studies, the presence of prey, or a prey-like stimulus, was required for neuronal stimulation to induce predatory behaviour. This suggests these pathways promote hunting by potentiating visuomotor activity, rather than directly inducing release of motor programmes downstream of sensory perception. Recently, a glutamatergic pathway from SC to the subthalamic zona incerta (ZI) (*Shang et al., 2019*) and a GABAergic pathway from medial ZI to PAG (*Zhao et al., 2019*) have been shown to respond to prey-like sensory cues and promote hunting when optogenetically stimulated. However, optogenetic stimulation evokes limited expression of a subset of hunting-related behaviour in the absence of prey. The ZI→PAG pathway appears to integrate multiple visual, vibrissal and auditory prey-related sensory signals, discriminate prey from non-prey objects, and generate a positive valence signal to motivate predation (*Zhao et al., 2019*). Comparing and contrasting the properties of hunting-related circuits identified in different vertebrates may prove a useful means to characterise the neural substrates involved in detecting and attending to prey, motivating hunting behaviour, and controlling and sustaining (*Henriques et al., 2019*) predatory action sequences.

Notably, neurons that control the release of behavioural programmes, such as the *KalTA4u508* pretectal cells described here, represent key circuit loci at which sensory information might be integrated with internal state signals (*Flood et al., 2013*). In the context of vertebrate hunting, it will be important to elucidate the circuit mechanisms that modulate recruitment probability of such decision-making circuits, enabling animals to vary the expression of predatory behaviour in accordance with internal physiological states, experience, and competing behavioural demands.

## Materials and methods

**Key resources table**

| Reagent type (species) or resource | Designation | Source or reference | Identifiers | Additional information |
|---|---|---|---|---|
| Genetic reagent (*Danio rerio*) | Tg(KalTA4u508) u508Tg | This study | | Transgene |

*Continued on next page*

*Continued*

| Reagent type (species) or resource | Designation | Source or reference | Identifiers | Additional information |
|---|---|---|---|---|
| Genetic reagent (*Danio rerio*) | Tg(UAS:jGCaMP7f) u341Tg | This study | | Transgene |
| Genetic reagent (*Danio rerio*) | Tg(UAS:CoChR-tdTomato)u332Tg | This study | ZFIN ID: ZDB-ALT-190226–4 | Transgene |
| Genetic reagent (*Danio rerio*) | Tg(elavl3:H2B-GCaMP6s)jf5Tg | PMID: 25068735 | ZFIN ID: ZDB-ALT-141023–2 | *Vladimirov et al., 2014* |
| Genetic reagent (*Danio rerio*) | Tg(atoh7:gapRFP) cu2Tg | PMID: 17147778 | ZFIN ID: ZDB-ALT-070129–2 | *Zolessi et al., 2006* |
| Genetic reagent (*Danio rerio*) | Tg(UAS:GCaMP6f, cryaa:mCherry) icm06Tg | PMID: 28623664 | ZFIN ID: ZDB-ALT-160119–5 | *Knafo et al. (2017)* |
| Genetic reagent (*Danio rerio*) | Tg(UAS-E1b:NfsB-mCherry)jh17Tg | PMID: 17335798 | ZFIN ID: ZDB-ALT-110222–4 | *Davison et al. (2007)* |
| Genetic reagent (*Danio rerio*) | TgBAC(slc17a6b:loxP-DsRed-loxP-GFP)nns14Tg | PMID: 21199937 | ZFIN ID: ZDB-ALT-110413–5 | *Koyama et al. (2011)* |
| Genetic reagent (*Danio rerio*) | Tg(UAS:RFP)tpl2Tg | PMID: 24179142 | ZFIN ID: ZDB-ALT-131119–25 | *Auer et al. (2014)* |
| Genetic reagent (*Danio rerio*) | Tg(atoh7:GFP)rw021Tg | PMID: 12702661 | ZFIN ID: ZDB-ALT-050627–2 | *Masai et al. (2003)* |
| Genetic reagent (*Danio rerio*) | Tg(Cau.Tuba1:c3paGFP)a7437Tg | PMID: 22704987 | ZFIN ID: ZDB-ALT-120919–1 | *Bianco et al. (2012)* |
| Genetic reagent (*Danio rerio*) | Tg(elavl3:ITETA-PTET:Cr.Cop4-YFP)fmi2Tg | PMID: 23641200 | ZFIN ID: ZDB-ALT-120209–3 | *Fajardo et al. (2013)* |
| Genetic reagent (*Danio rerio*) | Tg(gata2a:GFP)pku2Et | PMID: 18164283 | ZFIN ID: ZDB-ALT-080514–1 | *Wen et al., 2008* |
| Genetic reagent (*Danio rerio*) | atoh7[th241] | PMID: 11430806 | ZFIN ID: ZDB-ALT-980203–363 | *Kay et al. (2001)* |
| Recombinant DNA reagent | UAS:CoChR-tdTomato | This study | | Plasmid |
| Sequence-based reagent | CoChR_fw PCR primer | This study | | CTCAGCGTAAAGCCAC CATGCTGGGAAACG |
| Sequence-based reagent | CoChR_rev PCR primer | This study | | TACTACCGGTGCCGCCACTGT |
| Sequence-based reagent | CoChR_tdT_fw PCR primer | This study | | ACAGTGGCGGCACCGGTAGTA |
| Sequence-based reagent | tdT_rev PCR primer | This study | | CTAGTCTCGAGATCTCCATGTTTAC TTATACAGCTCATCCATGCC |
| Sequence-based reagent | UAS_jGCaMP7_fw PCR primer | This study | | CGTAAAGCCACCATGGGTTCTCATC |
| Sequence-based reagent | UAS_jGCaMP7_rev PCR primer | This study | | CTCGAGATCTCCATGTTTACT TCGCTGTCATCATTTGTACAAAC |
| Sequence-based reagent | pvalb_fw PCR primer | This study | | GGGGACAAGTTTGTACAAAAAAGC AGGCTGGATGGTGGGCCAAATCAAAGGCTAC |
| Sequence-based reagent | pvalb_rev PCR primer | This study | | GGGGACCACTTTGTACAAGAAAG CTGGGTGGAACGAGACCGGCAACACACAG |
| Antibody | Rabbit polyclonal anti-GFP; TP401 | AMS Biotechnology | RRID: AB_10890443 | 1:1000 |
| Antibody | Mouse monoclonal anti-44/42 MAPK (Erk1/2); 4696 | Cell Signaling Technology | RRID: AB_390780 | 1:500 |
| Antibody | Rat monoclonal anti-GFP; 04404–26 | Nacalai Tesque | RRID: AB_10013361 | 1:1000 |

*Continued on next page*

*Continued*

| Reagent type (species) or resource | Designation | Source or reference | Identifiers | Additional information |
|---|---|---|---|---|
| Antibody | Rabbit polyclonal anti-RFP/DsRed; PM005 | MBL International | RRID: AB_591279 | 1:1000 |
| Software, algorithm | MATLAB | MathWorks | RRID: SCR_001622 | https://uk.mathworks.com/products/matlab.html |
| Software, algorithm | LabView | National Instruments | RRID: SCR_014325 | http://www.ni.com/en-gb/shop/labview.html |
| Software, algorithm | Prism | GraphPad | RRID: SCR_002798 | https://www.graphpad.com/scientific-software/prism/ |
| Software, algorithm | FIJI | PMID: 22743772 | RRID: SCR_002285 | https://imagej.net/Fiji/Downloads |
| Software, algorithm | Simple Neurite Tracer (ImageJ plugin) | PMID: 21727141 | RRID: SCR_016566 | https://imagej.net/Simple_Neurite_Tracer |
| Software, algorithm | Advanced Normalization Tools (ANTs) | PMID: 20851191 | RRID: SCR_004757 | http://stnava.github.io/ANTs/ |
| Software, algorithm | Psychophysics Toolbox | PMID: 9176952 | RRID: SCR_002881 | http://psychtoolbox.org/ |
| Software, algorithm | Zebrafish Brain Browser (ZBB) | PMID: 26635538 | | https://science.nichd.nih.gov/confluence/display/burgess/Brain+Browser |
| Software, algorithm | MATLAB script for cell detection | PMID: 27881303 | | https://github.com/ahrens-lab/Kawashima_et_al_Cell_2016/ |
| Software, algorithm | Kernel Density Estimation Toolbox for MATLAB | https://www.ics.uci.edu/~ihler/code/kde.html | | |

## Experimental model and transgenic lines

Animals were reared on a 14/10 hr light/dark cycle at 28.5°C. For all experiments, we used zebrafish larvae homozygous for the *mitfa*$^{w2}$ skin-pigmentation mutation (*Lister et al., 1999*). Larvae used for pan-neuronal Ca$^{2+}$ imaging experiments were double-transgenic *Tg(elavl3:H2B-GCaMP6s)*$^{jf5Tg}$ (*Vladimirov et al., 2014*) and *Tg(atoh7:gapRFP)*$^{cu2Tg}$ (*Zolessi et al., 2006*). For AF7-pretectal Ca$^{2+}$ imaging, larvae were double-transgenic for *Tg(−2.5pvalb6:KalTA4)*$^{u508Tg}$ [*i.e. Tg(KalTA4u508)*; generated in this study, see below] and either *Tg(UAS:GCaMP6f,cryaa:mCherry)*$^{icm06Tg}$ (*Knafo et al., 2017*) or *Tg(UAS:jGCaMP7f)*$^{u341Tg}$ (generated in this study, see below). Larvae used to determine whether *KalTA4u508*-expressing neurons reside in AF7-pretectum were triple-transgenic *Tg(KalTA4u508)*, *Tg(UAS-E1b:NfsB-mCherry)*$^{jh17Tg}$ (*Davison et al., 2007*) and *TgBAC(slc17a6b:loxP-DsRed-loxP-GFP)*$^{nns14Tg}$ (*Koyama et al., 2011*). Larvae used for AF7 dendritic stratification analyses were triple-transgenic *Tg(KalTA4u508)*, *Tg(UAS:RFP)*$^{tpl2Tg}$ (*Auer et al., 2014*), and *Tg(atoh7:GFP)*$^{rw021Tg}$ (*Masai et al., 2003*). Fish used for mapping of cell location in the adult pretectum were triple-transgenic *Tg(KalTA4u508)*, *Tg(UAS:GCaMP6f,cryaa:mCherry)*$^{icm06Tg}$ and *Tg(atoh7:gapRFP)*$^{cu2Tg}$. Larvae used for photo-activatable GFP labelling were *Tg(Cau.Tuba1:c3paGFP)*$^{a7437Tg}$ (*Bianco et al., 2012*). Larvae used for single cell labelling and optogenetic stimulation of AF7-pretectal cells were double-transgenic *Tg(KalTA4u508)* and *Tg(elavl3:H2B-GCaMP6s)*$^{jf5Tg}$. Larvae used for single-cell optogenetic stimulation of AF7-pretectal cells in blind fish were double-transgenic *Tg(KalTA4u508)* and *Tg(atoh7:gapRFP)*$^{cu2Tg}$ with homozygous mutation of the *atoh7*$^{th241}$ gene (*Kay et al., 2001*). Blind *atoh7*$^{th241}$ homozygous fish were selected based on *Tg(atoh7:gapRFP)*$^{cu2Tg}$ expression being visible only in the eye but with no RGC projections in the brain. Larvae used for patterned optogenetic stimulation of the AF7-pretectal population were double-transgenic *Tg(KalTA4u508)* and *Tg(UAS:CoChR-tdTomato)*$^{u332Tg}$ (generated in this study, see below). Larvae used for pretectal cell ablations and free-swimming behaviour analyses were triple-transgenic *Tg(KalTA4u508)*, *Tg(UAS-E1b:NfsB-mCherry)*$^{jh17Tg}$ (*Davison et al., 2007*) and *Tg(elavl3:ITETA-PTET:Cr.Cop4-YFP)*$^{fmi2Tg}$ (*Fajardo et al., 2013*). Larvae used for assessment of AF7 axonal arborisations following ablation of

*KalTA4u508* pretectal neurons were triple-transgenic *Tg(KalTA4u508)*, *Tg(UAS-E1b:NfsB-mCherry)*[jh17Tg] and *Tg(atoh7:GFP)*[rw021Tg]. Larvae used for sham ablations of thalamic cells were double-transgenic *Tg(gata2a:GFP)*[pku2Et] (*Wen et al., 2008*) and *Tg(atoh7:GFP)*[rw021Tg]. Larvae used for optogenetic stimulation of the avOT were double-transgenic *Tg(atoh7:gapRFP)*[cu2Tg] and *Tg(elavl3:ITETA-PTET:Cr.Cop4-YFP)*[fmi2Tg] with either homozygous, heterozygous or no mutation of the *atoh7*[th241] gene. All larvae were fed *Paramecia* from 4 dpf onward. Animal handling and experimental procedures were approved by the UCL Animal Welfare Ethical Review Body and the UK Home Office under the Animal (Scientific Procedures) Act 1986.

## 2-photon calcium imaging and behavioural tracking

The procedure was very similar to that described in *Bianco and Engert (2015)*. Larval zebrafish were mounted in 3% low-melting point agarose (Sigma-Aldrich) at 5 dpf or 6 dpf and allowed to recover overnight before functional imaging at 6 dpf or 7 dpf. Imaging was performed using a custom-built 2-photon microscope [XLUMPLFLN 20 × 1.0 NA objective (Olympus), 580 nm PMT dichroic, band-pass filters: 510/84 (green), 641/75 (red) (Semrock), R10699 PMT (Hammamatsu Photonics), Chameleon II ultrafast laser (Coherent Inc)]. Imaging was performed at 920 nm with average laser power at sample of 5–10 mW. For imaging of *Tg(elavl3:H2B-GCaMP6s)* larvae, images (500 × 500 pixels, 0.61 µm/px) were acquired by frame scanning at 3.6 Hz and for each larva 10–14 focal planes were acquired with a z-spacing of 8 µm. For imaging of *Tg(KalTA4u508;UAS:GCaMP6f)* or *Tg(KalTA4u508;UAS:jGCaMP7f)* larvae, the same image size and scanning rate were used but 5–6 focal planes with a z-spacing of 5 µm were acquired for each larva. Visual stimuli were back-projected (Optoma ML750ST) onto a curved screen placed in front of the animal at a viewing distance of ~7 mm while a second projector provided constant background illumination below the fish. A coloured Wratten filter (Kodak, no. 29) was placed in front of both projectors to block green light from the PMT. Visual stimuli were designed in MATLAB using Psychophysics toolbox (*Brainard, 1997*). For all experiments, stimuli were presented in a pseudo-random sequence with 30 s inter-stimulus interval. Stimuli comprised 5° or 16°, dark or bright spots moving at 30°/s either left→right or right→left across ~200° of frontal visual space. Bright/dark spots had Weber contrast of 1 /– 1, respectively. In addition, 3 s whole-field bright/dark flashes were presented.

Eye movements were tracked at 60 Hz under 720 nm illumination using a FL3-U3-13Y3M-C camera (Point Grey) that imaged through the microscope objective. Tail movements were imaged at 430 Hz under 850 nm illumination using a sub-stage GS3-U3-41C6NIR-C camera (Point Grey). Microscope control, stimulus presentation and behaviour tracking were implemented using LabVIEW (National Instruments) and MATLAB (MathWorks).

## Calcium imaging analysis

All calcium imaging data analysis was performed using MATLAB scripts. Motion correction of fluorescence imaging data was performed as per *Bianco and Engert (2015)*. Regions of interest (ROIs) corresponding to cell nuclei were extracted using the cell detection code from *Kawashima et al. (2016)*. The time-varying fluorescence signal *F(t)* for each cell was extracted by computing the mean value of all pixels within the corresponding ROI binary mask at each time-point (imaging frame). The proportional change in fluorescence ($\Delta F/F_0$) at time *t* was calculated as

$$\Delta F/F_0 = \frac{F(t) - F_0}{F_0}$$

where $F_0$ is a reference fluorescence value, taken as the median of *F(t)* during the 30 frames prior to all visual stimulus presentations.

We used multilinear regression to model the fluorescent timeseries of each imaged neuron (ROI) in terms of simultaneously recorded kinematic predictors ('regressors'). Regressors were generated for oculomotor and locomotor variables (7 eye and 3 tail kinematics, see *Supplementary file 1*) by convolving time-series vectors for the relevant kinematic with a calcium impulse response function [CIRF, approximated as the sum of a fast-rising exponential, tau 20 ms, and a slow-decaying exponential, tau 420 ms for GCaMP6f and jGCaMP7f or 3 s for H2B-GCaMP6s; (*Miri et al., 2011*). To account for delays between neural activity and behaviour and/or indicator dynamics, we time-shifted the regressor matrix relative to the fluorescent response variable so as to minimise the residual

squared error of an ordinary least squares regression model. We used elastic-net regularised regression to improve interpretability and prediction accuracy in the presence of multicollinearity between the regressors (*Zou and Hastie, 2005*). Elastic net models were fit using the 'glmnet' package for MATLAB (*Qian et al., 2013*) and hyperparameters controlling the ratio of L1 vs. L2 penalty (alpha) and the degree of regularisation (lambda) were selected to minimise ten-fold cross-validated squared error. Model coefficients were then used to construct visuomotor vectors.

Visuomotor vectors (VMVs) were generated for each neuron by concatenating (a) the integral of $\Delta F/F_0$ in response to each visual stimulus (12 s time window from stimulus onset, mean integral across stimulus presentations) for presentations in which no eye convergence was performed by the larva (components 1–10); (b) multilinear regression coefficients ($\beta$ values) for eye, tail and motion correction regressors (components 11–21). VMVs from all imaged neurons were assembled into a matrix and each component was normalised across cells by dividing each column by its standard deviation.

VMV clustering was performed using a two-step procedure. First, we performed hierarchical agglomerative clustering of VMVs using a correlation distance metric (*Bianco and Engert, 2015*). For this first step, we selected only neurons that either exhibited strong visual responses (specifically, the maximum value of components 1–10 had to be within the top 5th percentile of such values across all neurons) or was well modelled in terms of behavioural kinematics ($R^2$ had to be within the top 5th percentile of cross-validated $R^2$ across all neurons). The centroids of the 'seed' clusters generated in this step (correlation threshold, 0.7) constituted a set of archetypal response profiles. Next, the VMVs of the remaining neurons were assigned to the cluster with the closest centroid (within a correlation distance threshold of 0.7) to produce the final clusters.

To assign cluster identities to *KalTA4u508* pretectal neurons (*e.g. Figures 4A, 5T*), VMVs were generated as described above and the same assignment strategy and correlation distance threshold (0.7) were used. Note that normalisation of components was performed using the standard deviations computed for the initial matrix of VMVs.

Hunting Index (HIx) scores were calculated for each cell as follows. For each hunting response, convergence-triggered activity was measured by computing the mean of z-scored GCaMP fluorescence in a time window (±1 s) centred on the convergent saccade, $x_{Ri}$. Next, activity was measured at the same time during non-response trials in which the same visual stimulus was presented. The difference between $x_{Ri}$ and the mean of non-response activity, $\mu_{NR}$, was computed:

$$d_i = x_{Ri} - \mu_{NR}$$

HIx scores were computed as the mean of these $d_i$ distance values across all response trials during which the cell was imaged. HIx values were computed separately for convergence events paired with leftwards tail movements, rightwards tail movements, and symmetrical/no tail movements.

## 3D image registration

Registration of image volumes was performed using the ANTs toolbox version 2.1.0 (*Avants et al., 2011*). Images were converted to the NRRD file format required by ANTs using ImageJ. As an example, to register the 3D image volume in 'fish1_01.nrrd' to the reference brain 'ref.nrrd', the following parameters were used:

```
antsRegistration -d 3 -float 1 -o [fish1_, fish1_Warped.nii.gz] -n BSpline -r
[ref.nrrd, fish1_01.nrrd, 1] -t Rigid[0.1] -m GC[ref.nrrd, fish1_01.nrrd, 1, 32,
Regular, 0.25] -c [200 × 200×200 × 0,1e-8, 10] -f 12 × 8×4 × 2 s 4 × 3×2 × 1 t
Affine[0.1] -m GC[ref.nrrd, fish1_01.nrrd, 1, 32, Regular, 0.25] -c [200 ×
200×200 × 0,1e-8, 10] -f 12 × 8×4 × 2 s 4 × 3×2 × 1 t SyN[0.1, 6, 0] -m CC[ref.nrrd,
fish1_01.nrrd, 1, 2] -c [200 × 200×200x200 × 10,1e-7, 10] -f 12 × 8×4x2 × 1 s 4 ×
3×2x1 × 0
```

The deformation matrices computed above were then applied to any other image channel N of fish1 using:

```
antsApplyTransforms -d 3 v 0 -float -n BSpline -i fish1_01.nrrd -r ref.nrrd -o
fish1_0N_Warped.nii.gz -t fish1_1Warp.nii.gz -t fish1_0GenericAffine.mat
```

All brains were registered onto the ZBB brain atlas (1 × 1 × 1 xyz µm/px) (*Marquart et al., 2015*; *Marquart et al., 2017*) and onto a high-resolution *Tg(elavl3:H2B-GCaMP6s)* reference brain (0.76 × 0.76 × 1 xyz µm/px, mean of 3 larvae), with some differences between experiments:

- For functional calcium imaging volumes, a three-step registration was used: the imaging volume, composed of 10–14 image planes (500 × 500 px, 0.61 µm/px, 8 µm z-spacing), was first registered to a larger volume of the same brain acquired at the end of the experiment (1 µm z-spacing), using affine and warp transformations. Then, the larger volume was registered to the Hi-Res *Tg(elavl3:H2B-GCaMP6s)* reference brain. Because the high-resolution volume had already been registered onto the ZBB atlas, the transformations were concatenated to bring the functional imaging volume to the ZBB atlas (calcium imaging stack → post-imaging stack → Hi-Res → ZBB).
- The brain regions displayed in *Figure 3F,H*, and *Figure 3—figure supplement 1B,C* correspond to volumetric binary image masks in the ZBB atlas that have been registered to the Hi-Res *Tg(elavl3:H2B-GCaMP6s)* reference brain using the ZBB *Tg(elavl3:H2B-RFP)* volume and performing affine and warp transformations (ZBB *elavl3:H2B-RFP* → Hi-Res).
- For the registration displayed in *Figure 3A,B* of *KalTA4u508* neurons to the Hi-Res *Tg(elavl3: H2B-GCaMP6s)* reference brain, the imaging volume was registered to the ZBB *Tg(vglut: DsRed)* volume [previously registered to the Hi-Res reference brain (ZBB *elavl3:H2B-RFP* → Hi-Res] using the vglut2a channel [*TgBAC(slc17a6b:loxP-DsRed-loxP-GFP)*] acquired in parallel with *Tg(KalTA4u508;UAS-E1b:NfsB-mCherry)* imaging and performing affine and warp transformations.
- For single *KalTA4u508* neuron tracing experiments, the imaging volume was registered to the Hi-Res *Tg(elavl3:H2B-GCaMP6s)* reference brain using the H2B-GCaMP6s channel acquired in parallel with *Tg(KalTA4u508);UAS-CoChR-tdTomato-injected* imaging. Affine and warp transformations were performed to bring the imaging volume and associated neuron tracing (see below) to the Hi-Res reference brain.
- Imaging volumes related to photo-activation of PA-GFP were registered to a whole-brain reference from a 6 dpf *Tg(α-tubulin:C3PA-GFP)* larva in which no photo-activation was performed, using affine and warp transformations. The *Tg(α-tubulin:C3PA-GFP)* reference volume was then registered to the ZBB atlas. The photo-activation volume was transported to the Hi-Res *Tg(elavl3:H2B-GCaMP6s)* reference by concatenating the transformations (photo-activation stack → *Tg(α-tubulin:C3PA-GFP)* reference → ZBB → Hi-Res).
- For the images and analyses displayed in *Figure 7B,C* and *Figure 7—figure supplement 1A,B* of sighted and blind *Tg(elavl3:itTA;Ptet:ChR2-YFP;atoh7:gapRFP)* larvae, the imaging volume was registered to the ZBB *anti-tERK* volume using the anti-tERK immunostain channel and performing affine and warp transformations.

All registration steps were manually assessed for global and local alignment accuracy. All brain regions referred to in this paper correspond to the volumetric binary image masks in the ZBB atlas, with the exception of regions in the anterior-ventral optic tectum, AF7-pretectum, and cholinergic nucleus isthmi (*Henriques et al., 2019*). These image masks, in ZBB reference space, can be downloaded as *Supplementary file 2–4*.

## DNA cloning and transgenesis

To generate the *UAS:CoChR-tdTomato* DNA construct used for single cell labelling and optogenetic stimulations and for creating the *Tg(UAS:CoChR-tdTomato)^{u332Tg}* line, the coding sequence of the blue light-sensitive opsin CoChR (from *pAAV-Syn-CoChR-GFP*) and the red fluorescent protein tdTomato (from *pAAV-Syn-Chronos-tdTomato*) were cloned in frame into a UAS Tol1 backbone (*pT1U-ciMP*). The *pAAV-Syn-CoChR-GFP* and *pAAV-Syn-Chronos-tdTomato* plasmids were gifts from Edward Boyden (Addgene plasmid # 59070 and # 62726, respectively) (*Klapoetke et al., 2014*). The *pT1UciMP* plasmid was a gift from Harold Burgess (Addgene plasmid # 62215) (*Horstick et al., 2015*). The cloning was achieved using the In-Fusion HD Cloning Plus CE kit (Clontech) with the following primers:

- CoChR_fw, CTCAGCGTAAAGCCACCATGCTGGGAAACG
- CoChR_rev, TACTACCGGTGCCGCCACTGT
- CoChR_tdT_fw, ACAGTGGCGGCACCGGTAGTA
- tdT_rev, CTAGTCTCGAGATCTCCATGTTTACTTATACAGCTCATCCATGCC

To generate the *UAS:jGCaMP7f* DNA construct used for creating the *Tg(UAS:jGCaMP7f)^u341Tg* line, the coding sequence of the genetically encoded calcium indicator jGCaMP7f (from *pGP-CMV-jGCaMP7f*) was cloned into the *pT1UciMP* UAS Tol1 backbone. The *pGP-CMV-jGCaMP7f* plasmid was a gift from Douglas Kim (Addgene plasmid # 104483) (*Dana et al., 2019*). As above, the cloning was achieved using the In-Fusion HD Cloning Plus CE kit (Clontech) with the following primers:

- UAS_jGCaMP7_fw, CGTAAAGCCACCATGGGTTCTCATC
- UAS_jGCaMP7_rev, CTCGAGATCTCCATGTTTACTTCGCTGTCATCATTTGTACAAAC

To generate the *Tg(UAS:jGCaMP7f)* and the *Tg(UAS:CoChR-tdTomato)* lines, purified *UAS:jGCaMP7f* or *UAS:CoChR-tdTomato* DNA constructs (35 ng/µl) were co-injected with Tol1 transposase mRNA (80 ng/µl) into *Tg(KalTA4u508)* zebrafish embryos at the early one-cell stage. Transient expression, visible as jGCaMP7f or tdTomato fluorescence, was used to select injected embryos that were then raised to adulthood. *Tol1* transposase mRNA was prepared by in vitro transcription from NotI-linearised *pCS2-Tol1.zf1* plasmid using the SP6 transcription mMessage mMachine kit (Life Technologies). The *pCS2-Tol1.zf1* was a gift from Harold Burgess (Addgene plasmid # 61388) (*Horstick et al., 2015*). RNA was purified using the RNeasy MinElute Cleanup kit (Qiagen). Germ line transmission was identified by mating sexually mature adult fish to *mitfa^w2/w2* fish and, subsequently, examining their progeny for jGCaMP7f or tdTomato fluorescence. Positive embryos from a single fish were then raised to adulthood. Once this second generation of fish reached adulthood, positive embryos from a single 'founder' fish were again selected and raised to adulthood to establish stable *Tg(KalTA4u508;UAS:jGCaMP7f)* and *Tg(KalTA4u508;UAS:CoChR-tdTomato)* double-transgenic lines.

The *Tg(–2.5pvalb6:KalTA4)^u508Tg* [*i.e. Tg(KalTA4u508)*] line was isolated as follows. First, we used Gateway cloning (Invitrogen) to construct an expression vector in which ~ 2.5 kb of zebrafish genomic sequence upstream of the *pvalb6* gene start codon was placed upstream of the KalTA4 (*Distel et al., 2009*) open reading frame. The genomic sequence was cloned using the following primers and Phusion PCR polymerase (Thermo Fisher Scientific):

- pvalb_fw: GGGGACAAGTTTGTACAAAAAAGCAGGCTggatggtgggccaaatcaaaggctac
- pvalb_rev: GGGGACCACTTTGTACAAGAAAGCTGGGTggaacgagaccggcaacacacag

(where capital letters indicate the attB1/B2 extension sequences).

The expression vector was then micro-injected into one-cell stage *Tg(UAS-E1b:Kaede)^s1999t* (*Davison et al., 2007*) embryos at 30 ng/µl along with *tol2* mRNA (30 ng/µl) and adult fish were screened for germline transmission by outcrossing as described above. This expression vector generated a wide range of expression patterns, one of which was designated the allele *u508Tg* and labelled AF7-pretectal neurons, as well as numerous other neuronal populations in the brain, spinal cord and sensory ganglia as reported here.

## Immunohistochemistry

### Larvae

Samples were fixed overnight in 4% paraformaldehyde (PFA) in 0.1 M phosphate buffered saline (PBS, Sigma-Aldrich) and 4% sucrose (Sigma-Aldrich) at 4°C. Brains were manually dissected with forceps prior to immunostaning. First, dissected brains were permeabilised by incubation in proteinase-K (40 µg/ml) in PBS with 1% Triton-X100 (PBT, Sigma-Aldrich) for 15 min. This was followed by 3 × 5 min washes in PBT, 20 min fixation in 4% PFA at room temperature and 3 × 5 min washes in PBT. Second, brains were incubated in block solution (2% goat serum, 1% DMSO, 1% BSA in PBT, Sigma-Aldrich) for 2 hr. Subsequently, brains were incubated in block solution containing primary antibodies overnight, followed by 6 × 1 hr washes in PBT on a slowly rotating shaker. Third, brains were incubated in block solution containing secondary antibodies overnight, followed by 6 × 1 hr washes in PBT. Finally, PBT was rinsed out by doing washes in PBS and brains were stored at 4°C. Imaging was performed using the two-photon microscope described above at 790 nm. Primary antibodies were: rabbit anti-GFP (AMS Biotechnology, TP401, dilution 1:1000) and mouse anti-ERK (Cell Signaling Technology, 9102, p44/42 MAPK (Erk1/2), dilution 1:500). Secondary antibodies were: goat anti-rabbit Alexa Fluor 488-conjugated (Thermo Fisher Scientific, A-11034, dilution 1:200), and goat anti-mouse Alexa Fluor 594-conjugated (Thermo Fisher Scientific, A-11005, dilution 1:200).

## Adults

*Tg(KalTA4u508;UAS:GCaMP6f;atoh7:gapRFP)* fish (3 months old) were deeply anesthetised in 0.2% tricaine (MS222, Sigma) and fixed in 4% paraformaldehyde (PFA) for 24 hr at 4°C. Brains were then carefully dissected under a stereomicroscope and transferred to saline phosphate buffer (PBS), where they were maintained for at least half an hour. Two different procedures were used for sectioning the brains. For cryostat sectioning, brains were cryopreserved, embedded in Tissue Tek OCT compound (Sakura Finetek) and frozen using liquid nitrogen cooled methylbutane. Transverse sections of the brain (12 µm thick) were obtained using a cryostat and collected in gelatine-coated slides. For vibratome sectioning, brains were first embedded into 3% agarose in PBS. Transverse sections of the brains (100 µm thick) were obtained using a vibratome and transferred to PBS in microtubes. Immunostaining was performed by either adding solutions onto the cryostat sections or changing the solutions inside the microtubes. Sections were incubated first in normal goat serum (Sigma, dilution 1:10) in PBS with 0.5% Triton for 1 hr at room temperature, and then with a cocktail of two primary antibodies (rat anti-GFP, Nacalai Tesque, 04404–26, dilution 1:1000, and rabbit anti-RFP, MBL International, PM005, dilution 1:1000) for 24 hr at room temperature. Next, after three washes in PBS, brain sections were incubated with a cocktail of two secondary antibodies (goat anti-rat Alexa 488, Thermo Fisher Scientific, A-11006, dilution 1:500, and goat anti-rabbit Alexa 568, Thermo Fisher Scientific, A-11011, dilution 1:500) for 1 hr at room temperature. Sections were washed in PBS, mounted with 50% glycerol in PBS and imaged using a Nikon A1R confocal microscope equipped with a Nikon Plan Fluor 20 × 0.50 NA objective. Excitation light was provided by an argon ion multichannel laser tuned to 488 nm (green channel), and a 561 nm diode laser (red channel).

## Single cell labelling

To label individual *KalTA4u508* neurons, *UAS:CoChR-tdTomato* DNA constructs were injected into 1–4 cell stage *Tg(KalTA4u508)* or *Tg(KalTA4u508;atoh7:gapRFP)* embryos with homozygous or no mutation of the *atoh7$^{th241}$* gene. Plasmid DNA was purified using midi-prep kits (Qiagen) and injected at a concentration of 30 ng/µl in distilled water. Larvae (4 dpf) were then screened for CoChR-tdTomato expression. Only larvae showing expression in a single *KalTA4u508* pretectal neuron were subsequently used for optogenetic stimulations and neuronal tracing experiments. Single cell morphologies were traced using the Simple Neurite Tracer plugin for ImageJ (*Longair et al., 2011*).

## Anatomical analyses

Cell density of neurons belonging to hunting-initiation clusters (25–28) was computed in the following way. First, we obtained the soma 3D coordinates of all neurons in clusters 25–28 following anatomical registration to the high-resolution *Tg(elavl3:H2B-GCaMP6s)* reference brain. Then, we computed the local cell density at each soma location using an adaptive Gaussian-based kernel density estimate (*Breiman et al., 1977*), with the bandwidth at each point constrained to be proportional to the *k*th nearest neighbour distance where:

$$k = \sqrt{n}$$

and *n* is the number of neurons (N = 6,630 cells from 8 fish). To compute the kernel density estimate, we used a MATLAB-based toolbox developed by Alexander Ihler (www.ics.uci.edu/~ihler/code/kde.html). Images in Figure 3A,B represent volume projections in which hunting-initiation neurons are colour-coded according to local cell density.

Neurite stratification and axon projection profiles in *Figure 3D,G* were obtained by measuring fluorescence intensity values along the axes indicated on figure panels using ImageJ Line and Plot Profile tools. For each image channel, a maximum intensity projection image was generated before measuring fluorescence intensity. Each intensity profile *i* was then rescaled to generate a profile, *I*, ranging from 0 to 1 as follows:

$$I = \frac{i - i_{min}}{i_{max} - i_{min}}$$

where $i_{min}$ and $i_{max}$ are the minimum and maximum values of profile *i*, respectively.

## Photo-activation of PA-GFP

Larvae (5 dpf) homozygous for the *Tg(α-tubulin:C3PA-GFP)* transgene were anaesthetised and mounted in 2% low-melting temperature agarose. The same custom-built 2-photon microscope used for functional imaging was used to photo-activate PA-GFP in a small region (9 × 9 μm) containing cell bodies located in AF7-pretectum. The photo-activation site was selected by imaging the brain at 920 nm. Photo-activation was performed by continuously scanning at 790 nm (5 mW at sample) for 4 min. Larvae were then unmounted and allowed to recover. At 7 dpf, an image stack (1200 × 800 px, 0.38 μm/px, ~200 μm z-extent) was acquired at 920 nm covering a large portion of the midbrain, tegmentum and hindbrain. Axonal projections were traced using the Simple Neurite Tracer plugin for ImageJ (*Longair et al., 2011*).

## Monitoring of free-swimming behaviour

The same behavioural tracking system was used for both optogenetic stimulations and assessment of visuomotor behaviours with some differences. Images were acquired under 850 nm illumination using a high-speed camera [Mikrotron MC1362, 250 Hz (*optogenetic stimulations*) or 700 Hz (*visuomotor behaviours assessment*), 500 μs shutter-time] equipped with a machine vision lens (Fujinon HF35SA-1) and an 850 nm bandpass filter to block visible light. In all experiments, larvae were placed in the arena and allowed to acclimate for around 2 min before starting experiments.

### Optogenetic stimulations

Larvae were placed in a petri dish with a custom-made agarose well (28 mm diameter, 3 mm depth) filled with fish facility water. Blue light was delivered across the whole arena from above using a 470 nm LED (OSRAM Golden Dragon Plus, LB W5AM) or a 459 nm LED (OSRAM OSTAR Projection Power, LE B P2W). Irradiance (0.44–2.55 mW/mm$^2$) was varied using constant current drive electronics with pulse-width modulation at 5 kHz. 'LED-On' trials included 7 s periods of continuous blue light illumination, interleaved with 'no-stimulation' trials in which no blue light was provided. Both LED-On and no-stimulation trials lasted 8 s. A minimum of 10 LED-On trials were acquired for each fish. LED and camera control were implemented using LabVIEW (National Instruments).

### Assessment of visuomotor behaviours

Larvae were placed in a 35 mm petri dish filled with 3.5 ml of fish facility water. Visual stimuli were projected onto the arena from below using an AAXA P2 Jr Pico Projector via a cold mirror. Visual stimuli were designed using Psychophysics Toolbox (*Brainard, 1997*). Looming stimuli expanded from 10 to 100° with L/V ratio of 255 ms (*Dunn et al., 2016*). Optomotor gratings had a period of ~10 mm and moved at one cycle/s. Optomotor gratings and looming spots were presented in egocentric coordinates such that directional gratings always moved 90° to left or right sides with respect to fish orientation and looming spots were centred 5 mm away from the body centroid and at 90° to left or right. Stimuli were presented in pseudo-random order with an inter-stimulus interval of minimum 60 s. Stimuli were only presented if the body centroid was within a predefined central region (11 mm from the edge of the arena). If this was not the case, a concentric grating was presented that moved towards the centre of the arena to attract the fish to the central region. At the beginning of each experiment, 60 *Paramecia* were added to arena. Each experiment typically lasted ~1 hr. Final *Paramecia* numbers were counted manually from full-frame video data from the final 10 s of each experiment and adjusted for consumption in 60 min [multiplying by 60/experiment duration (min)]. During experiments, eye and tail kinematics were tracked online as described below. Camera control, online tracking and stimulus presentation were implemented using LabVIEW (National Instruments) and MATLAB (MathWorks).

## Analyses of free-swimming behaviour

Data analysis was performed using LabVIEW (National Instruments) and MATLAB (MathWorks). Eye and tail kinematics were tracked offline for optogenetic experiments, and online for assessment of visuomotor behaviours with some differences. First, images were background-subtracted using a background model generated over 8 s in which the larva was moving (*offline tracking*), or a continuously updated background model (*online tracking*). Next, images were thresholded and the body centroid was found by running a particle detection routine for binary objects within suitable area

limits. For online tracking, eye centroids were detected using a second threshold and particle detection procedure with the requirement that these centroids were in close proximity to the body centroid. For offline tracking, eye centroids were detected using a particle detection procedure that uses both binary and greyscale images to identify the two centroids within suitable area limits that had the lowest mean intensity values. For online tracking, body and eye orientations were computed using second- and third-order image moments. For offline tracking, body orientation was computed as the angle of the vector formed by the centre of mass of the body centroid (origin) and the mid-point between the eye centroids. Eye orientation was computed as the angle between the major axis of the eye and the body orientation vector. Vergence angle was computed as the difference between the left and right eye angles. The tail was tracked by performing consecutive annular line-scans, starting from the body centroid and progressing towards the tip of the tail so as to define nine equidistant x-y coordinates along the tail. Inter-segment angles were computed between the eight resulting segments. Reported tail curvature was computed as the sum of these inter-segment angles. Rightward bending of the tail is represented by positive angles and leftward bending by negative angles. To identify periods of high ocular vergence, which represent hunting routines, a vergence angle threshold was computed for each fish by fitting a two-term Gaussian model to its vergence angle distribution. A fish was considered to be hunting if vergence angle exceeded this vergence threshold. For experiments assessing visuomotor behaviours, the vergence angle distribution was invariably bimodal and the vergence threshold was computed as one standard deviation below the centre of the higher angle Gaussian. For optogenetic experiments, in cases where the vergence angle distribution was not bimodal, a fixed vergence threshold of 55° was used.

## Optogenetic experiments

Response probability was computed as the fraction of LED-On trials in which at least one hunting routine (*i.e.* period with ocular vergence above threshold) was detected during the 7 s stimulation period. Similarly, for no-stimulation trials, response probability is the fraction of trials in which at least one hunting-like routine was detected. Response latency for LED-On trials was calculated from light stimulus onset. Swim bouts were identified using velocity thresholds (800°/s for bout onset, 200°/s for bout offset) applied to smoothed absolute tail angular velocity traces. Tail beat frequency was computed as the reciprocal of the mean full-cycle period during a swim bout. Tail vigour is computed by integrating absolute tail angular velocity (smoothed with a 40 ms box-car filter) over the first 120 ms of a swim bout. Bout asymmetry measures the degree to which tail curvature during a bout shows the same laterality as that determined during the first half-beat. It is computed as the fraction of time points in which the sign of tail angle matches the direction of the first half beat. This metric is high for hunting related J-turns but close to zero for forward swims. For each bout, the fraction of total curvature localised to the distal third of the tail was computed for the first half beat. To identify capture swim-like movements during optogenetically evoked hunting routines, movies were individually inspected and the following criteria were used to classify a swim bout as capture swim: (1) small change in body orientation associated with (2) divergence of the eyes and (3) jaw movement/suction and/or (4) dorsiflexion of the body.

## Assessment of visuomotor behaviours

Escape responses to loom stimuli were identified if the instantaneous speed of the body centroid exceeded 75 mm/s. An optomotor response gradient [OMR turn rate (°/s)] was calculated for each presentation as the total change in orientation during the stimulus presentation divided by the duration of the presentation [for leftwards OMR stimuli, the OMR turn rate (typically negative) was multiplied by –1 to group the data with rightwards OMR stimuli]. Mean swim speed was calculated as the total distance covered by the larva in the central region of the arena divided by the total time spent in this region.

## Optogenetic stimulation of AF7-pretectum in tethered larvae

Patterned illumination was delivered to the pretectum of tethered larvae using a custom-built digital micromirror device (DMD) rig. The DMD (Texas Instruments DLP LightCrafter 6500) was illuminated using a liquid light-guide coupled 470 nm LED (Mightex BLS-LCS-0470-50-22). A relayed image of the DMD was expanded and projected onto the sample plane of the objective (Zeiss N-Achroplan

20 × 0.5 NA) such that individual micromirrors were 0.46 × 0.46 μm at sample. Larvae, mounted in the same way as for calcium imaging, were imaged at 100 Hz under 850 nm illumination using a sub-stage FL3-U3-13Y3M-C camera (Point Grey). Trials were 10 s long and included a 4 s period of continuous blue light illumination of either the left or right AF7-pretectum (~75 × 75 μm target, 22.1 mW/mm$^2$). A minimum of 6 stimulation trials were performed for each target region. Movies were individually inspected to identify hunting-like events, which were defined by execution of saccadic eye convergence. The apparatus was controlled using LabVIEW (National Instruments).

### Laser ablations

*KalTA4u508* pretectal neurons were targeted for ablation in 6 dpf *Tg(KalTA4u508;UAS:mCherry; elavl3:itTA;Ptet:ChR2-YFP)* or *Tg(KalTA4u508;UAS:mCherry;atoh7:GFP)* larvae, which were anaesthetised using MS222 and mounted in 1% low-melting temperature agarose (Sigma-Aldrich). Ablations were performed using a MicroPoint system (Andor) attached to a Zeiss Axioplan-2 microscope equipped with a Zeiss Achroplan water-immersion 63 × 0.95 NA objective. A pulsed nitrogen-pumped tunable dye laser (Coumarin-440 dye cell) was focused onto individual *KalTA4u508* neurons and pulses were delivered at a frequency of 10 Hz for 60–120 s. All visible *KalTA4u508* neurons in both hemispheres were targeted for ablation and cell damage was confirmed under DIC optics. Larvae were then unmounted and allowed to recover overnight. Sham ablations of thalamic neurons were performed in 6 dpf *Tg(gata2a:GFP;atoh7:GFP)* larvae in a similar manner. The number of thalamic cells targeted for ablation was equivalent to the number of targeted *KalTA4u508* pretectal neurons (10–16 per hemisphere). Non-ablated larvae were mounted in agarose and underwent the same manipulations except for laser-ablation. Pre- and post-ablation image stacks were acquired with a 2-photon microscope at 790 nm (800 × 800 px, 0.38 μm/px, ~40 μm z-extent). Cell counting was performed manually in ImageJ using the multi-point tool.

### Quantification and statistical analysis

Statistical analyses were performed in Prism 8 (GraphPad) and MATLAB R2017b (MathWorks). Statistical tests, p-values, N-values, and additional information are reported in *Supplementary file 1*. All tests were two-tailed and were chosen after data were tested for normality and homoscedasticity.

### Data/resource sharing

Data generated or analysed during this study are included in the manuscript and supporting files. Source data files have been provided for *Figures 1* and *2–8*.

## Acknowledgements

The authors thank members of the Bianco lab, David Attwell, Tiago Branco, Tara Keck and Steve Wilson for helpful discussions and critical feedback on the manuscript and UCL Fish Facility staff for fish care and husbandry. We thank Richard Poole and Arantza Barrios for help with the laser-ablation system and Claire Wyart for sharing the *Tg(UAS:GCaMP6f,cryaa:mCherry)$^{icm06Tg}$* line prior to publication. PA was supported by a Sir Henry Wellcome Postdoctoral Fellowship (204708/Z/16/Z). IHB was supported by a Sir Henry Dale Fellowship from the Royal Society and Wellcome Trust (101195/Z/13/Z) and a UCL Excellence Fellowship.

## Additional information

### Funding

| Funder | Grant reference number | Author |
| --- | --- | --- |
| Wellcome | Sir Henry Dale Fellowship, 101195/Z/13/Z | Isaac H Bianco |
| Wellcome | Sir Henry Wellcome Postdoctoral Fellowship, 204708/Z/16/Z | Paride Antinucci |
| University College London | Excellence Fellowship | Isaac H Bianco |

| Royal Society | Sir Henry Dale Fellowship, 101195/Z/13/Z | Isaac H Bianco |

The funders had no role in study design, data collection and interpretation, or the decision to submit the work for publication.

## Author contributions

Paride Antinucci, Conceptualization, Software, Formal analysis, Funding acquisition, Investigation, Visualization, Methodology, Writing—original draft, Writing—review and editing; Mónica Folgueira, Investigation, Methodology; Isaac H Bianco, Conceptualization, Resources, Software, Supervision, Funding acquisition, Methodology, Writing—original draft, Project administration, Writing—review and editing

## Author ORCIDs

Paride Antinucci https://orcid.org/0000-0003-0573-5383
Mónica Folgueira http://orcid.org/0000-0003-2927-7516
Isaac H Bianco https://orcid.org/0000-0002-3149-4862

## Ethics

Animal experimentation: Animal handling and experimental procedures were approved by the UCL Animal Welfare Ethical Review Body and the UK Home Office under the Animal (Scientific Procedures) Act 1986.

## Decision letter and Author response

Decision letter https://doi.org/10.7554/eLife.48114.037
Author response https://doi.org/10.7554/eLife.48114.038

# Additional files

## Supplementary files

• Supplementary file 1. Regression and statistical details. Spreadsheet containing details of statistical analyses (test used, test statistic, values of N, centre and spread), and description of kinematic regressors.
DOI: https://doi.org/10.7554/eLife.48114.031

• Supplementary file 2. Anatomical mask – avOT. TIFF stack containing binary mask defining the 'avOT' anatomical region, in ZBB space.
DOI: https://doi.org/10.7554/eLife.48114.032

• Supplementary file 3. Anatomical mask – AF7-pretectum. TIFF stack containing binary mask defining the 'AF7-pretectum' anatomical region, in ZBB space.
DOI: https://doi.org/10.7554/eLife.48114.033

• Supplementary file 4. Anatomical mask – NI chata. TIFF stack containing binary mask defining the 'NI chata' anatomical region, in ZBB space.
DOI: https://doi.org/10.7554/eLife.48114.034

• Transparent reporting form
DOI: https://doi.org/10.7554/eLife.48114.035

## Data availability

Data generated or analysed during this study are included in the manuscript and supporting files. Source data files have been provided for Figures 1 and 2—8.

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
