## [Decision Letter]

Thank you for submitting your article "A pretectal command system controls hunting behaviour" for consideration by *eLife*. Your article has been reviewed by three peer reviewers, including Vatsala Thirumalai as the Reviewing Editor and Reviewer #1, and the evaluation has been overseen by Ronald Calabrese as the Senior Editor.

The reviewers have discussed the reviews with one another and the Reviewing Editor has drafted this decision to help you prepare a revised submission.

Summary:

In this manuscript, the authors uncover the role of a group of pretectal neurons in generating hunting behaviour in larval zebrafish. By using an impressive combination of techniques and detailed functional, anatomical and behavioural analyses the authors have identified a brain region that contains a fraction of cells that can be stimulated to induce hunting-like behaviour and ablated to diminish the same behaviour. This work is technically impressive and the conceptual presentation is excellent. The activation of single neurons to induce hunting-like behaviour is already impressive, and the authors go beyond that by identifying fish that do show behavioural responses on activating a given cell, and image the same cell in that fish in a virtual behavioural assay to functionally characterize it. The reviewers suggest the following essential revisions to improve what is an already strong manuscript.

Essential revisions:

The authors make a strong statement that these neurons constitute a 'command system' for initiating hunting based on their finding that activation of single neurons could elicit hunting routines even in the absence of prey-like stimuli. However, the reviewers felt that such a strong claim was not sufficiently substantiated by the data presented for the following reasons:

1) Each neuron that was able to trigger prey capture did so only in a minority of trials (~18%) and did so with very long latencies (~4s). If they think that the optogenetic stimulation is triggering hunts by direct activation of AF7-pretec neurons, why is it successful only in a small percentage of trials and why does it take several seconds especially in the case of the neurons that project to the nMLF? Previous reports (Thiele et al., 2014; Del Maschio et al., 2017) suggest that direct activation of nMLF generates tail bends in less than a second after photostimulation. Even if AF7-pretec were 1-3 synapses away, delay seems longer than would be predicted for a command system. If this is likely due to low levels of stimulation of ChR2, did the authors test a range of light intensities? A likely alternative interpretation would be that AF7-pretec neurons encode "prey" value – activation of these neurons signal the existence of prey in the visual field, which then increases the probability of downstream generation of the hunting command.

2) The criterion for deciding that the stimulation induced a "hunting routine" is only observing a "period with ocular vergence above threshold". An alternative explanation would be that these neurons are premotor neurons driving eye vergence, and when the animal finds its eyes converged, it responds (with some delay) with a tail movement. Can the authors rule this out?

Given these concerns, the reviewers suggest that the authors reconsider their nomenclature of these neurons as a command system. The results are entirely consistent with the AF7-pretectal neurons playing a 'hunt-promoting' role. Indeed, this study parallels a recently published study (Zhao et al., 2019) identifying GABAergic neurons in zona incerta of mice as 'hunt-promoting' based on similar lines of experimentation. We believe that toning down on the "command system" hypothesis and instead positing this group of neurons as a "hunt-promoting" nucleus will be closer to the results and yet not take away from the importance of these findings.

In addition, reviewers have the following suggestions grouped broadly into experiments, analysis and presentation:

Experiments:

1) The authors stimulate single neurons using the *KalTA4u508* reagent. They do not report on the results of stimulating the entire population of AF7-pretectal neurons labeled by the *KalTA4u508* stable transgenic line. Is this because the line labels many neurons elsewhere in the brain? If not, it would be important to include this information. It is also essential to provide more information on the expression pattern in the *KalTA4u508* line across all regions of the brain and in the spinal cord.

2) Authors ablated these neurons to show effects on hunting. However, the control included no photo-ablation of any cells, and therefore does not control for damage to other nearby cells, or axons of RGCs. The authors should either perform a control with ablation of similar numbers of cells nearby, or justify the current control experiment in the text while acknowledging any shortcomings.

Analysis:

1) There is no 'Eye vergence only' regressor included in the analysis of activity. If the authors claim that eye convergence without tail movement never occurs then this might be important to show explicitly (again making clear what the delays are between the eye and tail movements).

2) Clustering: To obtain their functional clusters, the authors cluster the top 5% responsive neurons then assign the rest of the neurons to these clusters based on distance to cluster centroids. This analysis would need some additional information to make it more transparent. (a) What is the distribution between the cells selected with the two criteria, namely, high visually evoked activity or well modeled in terms of motor variables? (b) What numbers of cells formed the seeds for the 36 clusters (min-max range and median)? It is also recommended to add this criteria (that clustering was done on 5% of cells) in the Results text as well.

3) It was hard to be sure that the description of AF7 neurons scoring high on the "hunting index" (HIx) was a significant result as opposed to falling out of the nature of the regression. HIx measures predominance of activity during hunting as compared to non-response trials. But, in a sense, these are correlated: clustering picks out those cells having motor responses, hence likely to have a high HIx? What is the additional value of the HIx?

4) Figure 1E and F: In Bianco and Engert (2015), using the same moving spot paradigm the authors find that the "distribution of spot locations at time of convergent saccade did not differ for left-right versus right-left stimuli". In contrast this manuscript shows in Figure 1E and F a difference for left to right vs. right to left stimuli. Is this significant? What might account for this difference?

5) Cluster 26 and 28 show lateralized responses and in the fifth paragraph of the subsection “Pretectal neurons are recruited during hunting initiation”, authors state that their distributions are also lateralized but Figure 1—figure supplement 2D shows about equal numbers for cluster 26 on both sides. An explanation seems necessary.

Presentation:

1) The authors need to be clearer about the nature of the visuomotor vector that they use to regress Ca fluorescence. Is it possible to visualize an example VMV or two in the supplementary figures? The figures could be better explicated: for example, in Figure 1G, meaning of terms such as "Conv tail sym" are not immediately obvious until one gets fairly deep into the Materials and methods section.

2) To help the reader, the four tightly packed figures should be broken into perhaps 6 or 7. Currently, each figure is so dense that the font sizes are difficult to read. Further, it is harder to follow the logic when each figure is making multiple, often complex, points. Specifically, Figure 1J-O could become their own figure. The relevant subset of data shown in Figure 1—figure supplement 2A (i.e., the calcium responses during hunting and non-response for the key clusters of 1, 4, and 25-28) could be brought forward into that figure or another new figure. Figure 2 could likewise be split into anatomy of the AF7 neurons and their behavioral responses. The authors make a very nice analysis of the effect of the anterior ventral optic tectum (avOT) and retinal inputs on prey capture behavior. Unfortunately this has been placed in a supplementary figure (Figure 4—figure supplement 1). We would suggest that this be made into a figure on its own.

3) Subsection “Pretectal neurons labelled by *KalTA4u508* with hunting-initiation activity”, fourth paragraph: It's notable that the *KalTA4u508* neurons don't respond to visual stimuli alone, in marked contrast to the clusters of which they are members (cf. Figure 1N). Presumably this is because these neurons are a small subset of those clusters, but the authors should acknowledge this distinction in the Results and perhaps speculate on the implications – other cluster members must have more significant visual responses.

4) The Introduction and Discussion should be rewritten to position the AF7-pretectal neurons as a hunt-promoting nucleus instead of a command system. The title should also be suitably modified. It is best to avoid poorly defined terms such as 'behavioral epistasis' (in describing the interaction between avOT and AF7-pretec).

---

## [Author Response]

Essential revisions:The authors make a strong statement that these neurons constitute a 'command system' for initiating hunting based on their finding that activation of single neurons could elicit hunting routines even in the absence of prey-like stimuli. However, the reviewers felt that such a strong claim was not sufficiently substantiated by the data presented for the following reasons:1) Each neuron that was able to trigger prey capture did so only in a minority of trials (~18%) and did so with very long latencies (~4s). If they think that the optogenetic stimulation is triggering hunts by direct activation of AF7-pretec neurons, why is it successful only in a small percentage of trials and why does it take several seconds especially in the case of the neurons that project to the nMLF? Previous reports (Thiele et al., 2014; Del Maschio et al., 2017) suggest that direct activation of nMLF generates tail bends in less than a second after photostimulation. Even if AF7-pretec were 1-3 synapses away, delay seems longer than would be predicted for a command system. If this is likely due to low levels of stimulation of ChR2, did the authors test a range of light intensities? A likely alternative interpretation would be that AF7-pretec neurons encode "prey" value – activation of these neurons signal the existence of prey in the visual field, which then increases the probability of downstream generation of the hunting command.

We agree with the concerns about long latency and modest response probability observed with single neuron stimulation. Therefore, we have performed two additional experiments (which are now presented within Figure 5—figure supplement 1 and 2).

1) First, because the light intensities we could achieve for free swimming experiments were low (<1 mW/mm^2^) we tested whether response rates might increase (and latency decrease) at higher irradiance (using a new higher power LED enabling up to 2.5 mW/mm^2^). In addition, the observation that blind fish showed a substantial reduction in latency and increase in response rate in the context of avOT stimulation, suggested that perhaps the bright blue light interfered with induction of hunting. Therefore, we have also now generated and tested blind (*lakritz^-/-^*) fish expressing CoChR-tdTomato in single AF7-pretectal neurons. In one blind larva we found that response rate increased with irradiance, reaching 100% at 2.55 mW/mm^2^ whereas latency showed only a modest decrease at higher intensity. Notably, we could only generate a very small number of larvae that were both homozygous for the *lakritz* mutation, carriers of the necessary *KalTA4u508* and *atoh7:gapRFP* transgenes and had a single pretectal neuron labelled. Thus, these are preliminary observations, but we believe it is nonetheless informative to include this data within a figure supplement (Figure 5—figure supplement 1).

2) Second, we investigated the effect of stimulating multiple AF7-pretectal neurons. The *KalTA4u508* transgene indeed labels various other cell types (see below), preventing specific optogenetic targeting of pretectal neurons in free-swimming animals. Therefore, we used a patterned illumination system (a DMD rig) to focally stimulate the pretectal region in tethered *KalTA4u508;UAS:CoChR-tdTomato* larvae (Figure 5—figure supplement 2). In this experiment, where the full complement of *KalTA4u508*-labelled pretectal cells are stimulated and higher irradiance levels were possible (22 mW/mm^2^), we observed much higher hunting response probabilities and an order of magnitude reduction in response latency (to ~80 ms).

This new data indicates that the transgenically labelled *population* of AF7-pretectal cells can induce hunting with very short latency, compatible with a command system function. The long latencies observed with single cell stimulation might represent a limited capability of single pretectal neurons to bring downstream pattern generating circuits to threshold. We do not favour the interpretation that ‘AF7-pretectal neurons encode "prey" value – activation of these neurons signal the existence of prey in the visual field’ as *KalTA4u508* cells explicitly do not respond to the existence of prey but rather are recruited at initiation of hunting.

2) The criterion for deciding that the stimulation induced a "hunting routine" is only observing a "period with ocular vergence above threshold". An alternative explanation would be that these neurons are premotor neurons driving eye vergence, and when the animal finds its eyes converged, it responds (with some delay) with a tail movement. Can the authors rule this out?

It is correct that we use the period of elevated vergence to *detect* the occurrence and duration of (natural and optogenetically evoked) hunting routines. However, although this is our detection criterion, ocular vergence is not the only feature of those routines. In all fish where we could induce hunting, 100% of convergent saccades were paired with tail movement. This was true both in the free-swimming and tethered (DMD) optogenetic stimulation paradigms. The delay between saccadic convergence and tail movement averaged <10 ms, which is very similar to natural hunting behaviour. This data was already included in our first submission but for emphasis we have now included the eye-tail lag as an additional panel in the main figure. The tail movements that are coincident with saccadic convergence showed kinematic similarities with those observed during natural hunting (i.e. asymmetry, distal curvature, tail-beat frequency etc). Multiple swim bouts were typically observed during optogenetically evoked hunting (Figure 5—figure supplement 1B) and remarkably, the final swim bout sometimes resembled a capture swim, characterised by jaw-opening, dorsiflexion and eye divergence (observed for ~40% of cells and now quantified in Figure 5—figure supplement 1C, D). It is on this basis that we consider optogenetic stimulation of *KalTA4u508* cells induces hunting *routines*, rather than simply convergent saccades.

Given these concerns, the reviewers suggest that the authors reconsider their nomenclature of these neurons as a command system. The results are entirely consistent with the AF7-pretectal neurons playing a 'hunt-promoting' role. Indeed, this study parallels a recently published study (Zhao et al., 2019) identifying GABAergic neurons in zona incerta of mice as 'hunt-promoting' based on similar lines of experimentation. We believe that toning down on the "command system" hypothesis and instead positing this group of neurons as a "hunt-promoting" nucleus will be closer to the results and yet not take away from the importance of these findings.

The reviewers raised two valid concerns, but we believe that the new experimental data and analyses in this revised manuscript largely tackle both. Specifically, stimulating the *KalTA4u508* pretectal population triggers hunting with very short latency, equivalent to that observed for command neurons in other species (e.g. Inagaki et al., 2014). Second, evoked responses are not limited to saccadic convergence but comprise a sequence of oculomotor and locomotor actions with kinematic signatures of natural hunting behaviour. In the revised Discussion we now discuss the recent findings related to the SC–ZI–PAG pathway, published while this manuscript was under review. In distinction to AF7-pretectal neurons, ZI cells show *sensory* responses to prey-like features, appear to generate a positive valence signal to motivate predation and evoke limited hunting-like behaviour in absence of prey. This is quite distinct to the properties of *KalTA4u508* cells in AF7-pretectum. Moreover, *KalTA4u508* cells satisfy the criteria for command system neurons as defined by Yoshihara and Yoshihara (2018), as summarised in the Discussion.

However, in consideration of the reviewers’ comments we have made the following changes:

i) We have changed the title to ‘Pretectal cells control hunting behaviour’ and edited the Abstract to make clear that a command system function is our interpretation of the role of the cells ‘We propose that…’.

ii) We have removed all comments concerning command neurons from the Introduction and Results.

iii) We have edited the Discussion to further clarify that a command system role is a proposal based upon the experimental results. This is mostly confined to one section entitled ‘Do AF7-pretectal neurons comprise a command system controlling predation?’ which also includes discussion of the new experimental data and further clarification.

We hope that the reviewers consider these revisions accurately represent the experimental data and make explicit the basis for our interpretation.

In addition, reviewers have the following suggestions grouped broadly into experiments, analysis and presentation:Experiments:1) The authors stimulate single neurons using the KalTA4u508 reagent. They do not report on the results of stimulating the entire population of AF7-pretectal neurons labeled by the KalTA4u508 stable transgenic line. Is this because the line labels many neurons elsewhere in the brain? If not, it would be important to include this information. It is also essential to provide more information on the expression pattern in the KalTA4u508 line across all regions of the brain and in the spinal cord.

We should have made this clearer in the original manuscript. Indeed, the *KalTA4u508* transgene labels various additional cell populations within brain, spinal cord and sensory ganglia (e.g. lateral line), as well as some expression in muscle and skin. We now state this in the Results and explain that it is for this reason that we adopted a transient expression strategy to allow optogenetic stimulation specifically of the AF7-pretectal neurons. We also include an image of an entire transgenic larva (Figure 3—figure supplement 1A). The results of stimulating the entire (transgenically labelled) population of pretectal neurons, using the DMD set-up, is now presented (see above).

2) Authors ablated these neurons to show effects on hunting. However, the control included no photo-ablation of any cells, and therefore does not control for damage to other nearby cells, or axons of RGCs. The authors should either perform a control with ablation of similar numbers of cells nearby, or justify the current control experiment in the text while acknowledging any shortcomings.

We have now performed two ablation controls (see Figure 6—figure supplement 1). First, we ablated AF7-pretectal cells in larvae carrying the *atoh7:GFP* transgene to visualise RGC axons in AF7. These appeared undamaged by the ablation. Second, we ablated an equivalent number of neuronal somata slightly medial to the AF7-pretectal region (labelled by *gata2a:GFP*) and observed no impairment of hunting performance. These controls indicate our ablations were well targeted and cause impairment of hunting due to loss of *KalTA4u508* cells.

Analysis:1) There is no 'Eye vergence only' regressor included in the analysis of activity. If the authors claim that eye convergence without tail movement never occurs then this might be important to show explicitly (again making clear what the delays are between the eye and tail movements).

Eye convergence without tail movement is very rare, occurring for <10% of saccades. We now show this data in Figure 1—figure supplement 1 along with the eye-tail time lag during the virtual hunting assay (~10 ms). Due to the very limited number of events we did not generate an ‘eye vergence only’ regressor.

2) Clustering: To obtain their functional clusters, the authors cluster the top 5% responsive neurons then assign the rest of the neurons to these clusters based on distance to cluster centroids. This analysis would need some additional information to make it more transparent. (a) What is the distribution between the cells selected with the two criteria, namely, high visually evoked activity or well modeled in terms of motor variables? (b) What numbers of cells formed the seeds for the 36 clusters (min-max range and median?. It is also recommended to add this criteria (that clustering was done on 5% of cells) in the Results text as well.

The distributions of max visual response and regression goodness-of-fit (i.e. the metrics used for cell selection) are now shown in Figure 1—figure supplement 1D. The total fraction of cells used for clustering was 9.5% (top 5^th^ percentile for either visual or motor components). The numbers of cells in all of the seeds (as well as the final clusters) has been added as Figure 1—figure supplement 1F and G and the Results text has been edited as suggested.

3) It was hard to be sure that the description of AF7 neurons scoring high on the "hunting index" (HIx) was a significant result as opposed to falling out of the nature of the regression. HIx measures predominance of activity during hunting as compared to non-response trials. But, in a sense, these are correlated: clustering picks out those cells having motor responses, hence likely to have a high HIx? What is the additional value of the HIx?

The HIx result does not necessarily follow from the nature of the regression but is expected based on the results of the clustering. Because convergent saccades are correlated with presentation of prey-like stimuli, it is possible that a cell may have a significant ‘motor’ coefficient for convergence but in fact be encoding the presence of prey. The results of the clustering indicate this is, however, unlikely as the convergent saccade (hunt initiation) clusters have very minimal visual activity (extracted from non-response trials). The HIx analysis confirms this and isolates the modulation attributable to hunt initiation (which is not explicitly represented by any single VMV element). We believe this analysis (as well as the stimulus and motor-triggered calcium profiles) show in a very direct and convincing way that hunt initiation neurons are specifically activated in response trials, when the fish release convergent saccades.

4) Figure 1E and F: In Bianco and Engert (2015), using the same moving spot paradigm the authors find that the "distribution of spot locations at time of convergent saccade did not differ for left-right versus right-left stimuli". In contrast this manuscript shows in Figure 1E and F a difference for left to right vs. right to left stimuli. Is this significant? What might account for this difference?

The difference versus Bianco and Engert (2015) is likely due to pooling of locations for fast (30°/s) and slow (15°/s) stimuli in that prior study. Using exclusively 30°/s prey-like cues in this study, we consistently see a preference for the larvae to respond once the spot has crossed the midline and is moving nose–tail. Cumulative distribution functions with p-value have now been added in Figure 1—figure supplement 1.

5) Cluster 26 and 28 show lateralized responses and in the fifth paragraph of the subsection “Pretectal neurons are recruited during hunting initiation”, authors state that their distributions are also lateralized but Figure 1—figure supplement 2D shows about equal numbers for cluster 26 on both sides. An explanation seems necessary.

Figure 1—figure supplement 2D (now Figure 2—figure supplement 2C) shows data for *all* clustered neurons, whereas the statement in the Results text refers to cluster 26/28 cells *in AF7-pretectum*. The strongly lateralised anatomical distributions of these cells *in pretectum* is shown in the insets in Figure 2A.

Presentation:1) The authors need to be clearer about the nature of the visuomotor vector that they use to regress Ca fluorescence. Is it possible to visualize an example VMV or two in the supplementary figures? The figures could be better explicated: for example, in Figure 1G, meaning of terms such as "Conv tail sym" are not immediately obvious until one gets fairly deep into the Materials and methods section.

VMVs for all clustered neurons are displayed in Figure 1; these are not used to regress Ca fluorescence but instead describe the visual responses and regression coefficients. We have edited the schematic in Figure 1G to make this clearer. We suspect the reviewers are asking for examples of kinematic regressors alongside Ca data. This has now been added in Figure 1—figure supplement 1E. We have added a description of the Conv tail L/R/sym regressors/coefficients to the Results text to aid the reader.

2) To help the reader, the four tightly packed figures should be broken into perhaps 6 or 7. Currently, each figure is so dense that the font sizes are difficult to read. Further, it is harder to follow the logic when each figure is making multiple, often complex, points. Specifically, Figure 1J-O could become their own figure. The relevant subset of data shown in Figure 1—figure supplement 2A (i.e., the calcium responses during hunting and non-response for the key clusters of 1, 4, and 25-28) could be brought forward into that figure or another new figure. Figure 2 could likewise be split into anatomy of the AF7 neurons and their behavioral responses. The authors make a very nice analysis of the effect of the anterior ventral optic tectum (avOT) and retinal inputs on prey capture behavior. Unfortunately this has been placed in a supplementary figure (Figure 4—figure supplement 1). We would suggest that this be made into a figure on its own.

In this revised manuscript we now present the results across 8 main figures. We opted to split Figure 1 after panel K so that this figure now covers functional identification of hunting-related neurons, while localisation of hunting-initiation cells to the AF7-pretectal region is now presented separately in Figure 2. As suggested, the anatomy and response properties of *KalTA4u508* cells is now divided (into Figures 3 and 4). We have added an additional supplementary figure to present the DMD stimulation data and in line with the reviewers’ suggestion, now present the avOT data in a main figure, Figure 7. Figure 8 presents the effect of *KalTA4u508* ablation on avOT-induced hunting and the final circuit model. We have increased font sizes and hope that the new layout has improved presentation of the data.

3) Subsection “Pretectal neurons labelled by KalTA4u508 with hunting-initiation activity”, fourth paragraph: It's notable that the KalTA4u508 neurons don't respond to visual stimuli alone, in marked contrast to the clusters of which they are members (cf. Figure 1N). Presumably this is because these neurons are a small subset of those clusters, but the authors should acknowledge this distinction in the Results and perhaps speculate on the implications – other cluster members must have more significant visual responses.

We have added the following statement to the Results: ‘The absence of visual activity in *KalTA4u508* neurons is in contrast to the small responses seen previously in hunting-initiation clusters (Figure 2), suggesting some functional heterogeneity in clusters derived from pan-neuronal imaging.’

4) The Introduction and Discussion should be rewritten to position the AF7-pretectal neurons as a hunt-promoting nucleus instead of a command system. The title should also be suitably modified. It is best to avoid poorly defined terms such as 'behavioral epistasis' (in describing the interaction between avOT and AF7-pretec).

As discussed above, the Introduction, Discussion and title have been modified. The term ‘behavioural epistasis’ has been removed.